# Mesenchyme-derived IGF2 is a major paracrine regulator of pancreatic growth and function

**Constanze M. Hammerle**[1,2☯¤a], **Ionel Sandovici**[1,2,3☯], **Gemma V. Brierley**[1], **Nicola M. Smith**[1,2], **Warren E. Zimmer**[4], **Ilona Zvetkova**[1], **Haydn M. Prosser**[5¤b], **Yoichi Sekita**[1¤c], **Brian Y. H. Lam**[1], **Marcella Ma**[1], **Wendy N. Cooper**[1,2], **Antonio Vidal-Puig**[1], **Susan E. Ozanne**[1], **Gema Medina-Gómez**[6], **Miguel Constância**[1,2,3]*

**1** University of Cambridge Metabolic Research Laboratories and MRC Metabolic Diseases Unit, Institute of Metabolic Science, Addenbrookes Hospital, Cambridge, United Kingdom, **2** Department of Obstetrics and Gynaecology and National Institute for Health Research Cambridge Biomedical Research Centre, Cambridge, United Kingdom, **3** Centre for Trophoblast Research, Department of Physiology, Development and Neuroscience, University of Cambridge, Cambridge, United Kingdom, **4** Department of Medical Physiology, Texas A&M Health Science Center, College Station, Texas, United States of America, **5** The Wellcome Trust Sanger Institute, Genome Campus, Hinxton, United Kingdom, **6** Área de Bioquímica y Biología Molecular, Departamento de Ciencias Básicas de la Salud, Universidad Rey Juan Carlos, 28922-Alcorcón, Madrid, Spain

☯ These authors contributed equally to this work.
¤a Current address: Novo Nordisk A/S, Bagsværd, Denmark
¤b Current address: Cambridge Institute of Therapeutic Immunology & Infectious Disease. Jeffrey Cheah Biomedical Centre, Cambridge Biomedical Campus, University of Cambridge, United Kingdom
¤c Current address: Laboratory of Stem Cell Biology, Department of Biosciences, Kitasato University School of Science, Kanagawa, Japan
* jmasmc2@cam.ac.uk

**Data Availability Statement:** RNA-seq data have been deposited in the Gene Expression Omnibus (GEO) under the accession number GSE100981

## Abstract

The genetic mechanisms that determine the size of the adult pancreas are poorly understood. Imprinted genes, which are expressed in a parent-of-origin-specific manner, are known to have important roles in development, growth and metabolism. However, our knowledge regarding their roles in the control of pancreatic growth and function remains limited. Here we show that many imprinted genes are highly expressed in pancreatic mesenchyme-derived cells and explore the role of the paternally-expressed insulin-like growth factor 2 (*Igf2*) gene in mesenchymal and epithelial pancreatic lineages using a newly developed conditional *Igf2* mouse model. Mesenchyme-specific *Igf2* deletion results in acinar and beta-cell hypoplasia, postnatal whole-body growth restriction and maternal glucose intolerance during pregnancy, suggesting that the mesenchyme is a developmental reservoir of IGF2 used for paracrine signalling. The unique actions of mesenchymal IGF2 are demonstrated by the absence of any discernible growth or functional phenotypes upon *Igf2* deletion in the developing pancreatic epithelium. Additionally, increased IGF2 levels specifically in the mesenchyme, through conditional *Igf2* loss-of-imprinting or *Igf2r* deletion, leads to pancreatic acinar overgrowth. Furthermore, *ex-vivo* exposure of primary acinar cells to exogenous IGF2 activates AKT, a key signalling node, and increases their number and amylase production. Based on these findings, we propose that mesenchymal *Igf2*, and perhaps other imprinted genes, are key developmental regulators of adult pancreas size and function.

**Funding:** This work was supported by Biotechnology and Biological Sciences Research Council (grant BB/H003312/1 to M.C., S.E.O., A.V. P.); the Medical Research Council ([MRC_MC_UU_12012/4 and MRC_MC_UU_00014/4] to M.C and S.E.O, [MRC 979241] to C.H., [MRC_MC_UU_12012/5] to Metabolic Diseases Unit), the Wellcome Trust ([Strategic Award 100574/Z/12/Z], [102355/Z/13/Z] to N.M.S.) and [098051] for miR-483 knockout production), and the Spanish Ministry of Economy and Competitiveness (grants BFU2012-33594 and BFU2013-47384-R to G. M.-G.). The funders had no role in study design, data collection and analysis, decision to publish, or preparation of the manuscript.

**Competing interests:** The authors have declared that no competing interests exist.

## Author summary

The pancreas is formed of two main components: the exocrine pancreas (producing digestive enzymes that break down food so it can be easily absorbed by the intestine) and the endocrine pancreas (producing insulin and other hormones that control blood sugar levels). Additionally, the pancreas contains stromal cells (mesenchyme-derived cells) that support the function of the exocrine and endocrine components. We know little about how the pancreas reaches its normal size. In this study, using mouse genetic engeneering, we explored the roles played by a hormone-like gene called *Igf2*, that is similar in structure to insulin, and is active only on the chromosome inherited from the father. We found that within the pancreas, *Igf2* is mostly active in the mesenchyme-derived cells. When *Igf2* is lost specifically within these cells, the entire pancreas becomes smaller, with reduced capacity to produce digestive enzymes and to maintain normal blood sugar levels during pregnancy. Increased IGF2 levels in the mesenchyme-derived cells leads to a larger pancreas, while no effects are observed when *Igf2* is lost in the exocrine and endocrine pancreas. Our results demonstrate that *Igf2* activity in mesenchyme-derived cells is key for the control of pancreas size and function.

## Introduction

The mammalian pancreas plays a central role in energy homeostasis, which is achieved by functionally and morphologically distinct exocrine and endocrine components. Optimal pancreatic function requires a match between pancreas size and the physiological demands of the host organism. However, the genetic determinants of organ size, and the mechanisms that achieve adequate relative organ size, are poorly understood. The size of the pancreas is thought to be fixed early in development, limited by the size of the progenitor cell pool that is established in the developing pancreatic bud [1]. In addition to autonomous cues in the epithelium, pancreatic development is also dictated by non-autonomous signals from the mesenchyme. Mesenchymal cells overlie the developing pancreatic bud and provide critical signals for the expansion of both precursors and differentiated endocrine and exocrine cells [2]. These cells are present throughout pancreas organogenesis, but the relative proportion of mesenchyme to epithelium changes, with a dramatic reduction over time, as the mesenchyme differentiates into more specialized cell types such as pericytes [3] and epithelial cells expand. Recent elegant genetic manipulation approaches in the mouse have shown that mesenchymal cells regulate pancreatic growth and branching at both early and late *in utero* developmental stages [2,4]. However, many of the mesenchymal signals that control these processes remain un-identified. Moreover, the factors required for cell fate decisions that specify individual pancreas cell type subsets are fairly well established but less is known about signalling pathways involved in proliferation and survival of cell types.

Insulin-like growth factors (IGF1 and IGF2) are small mitogenic polypeptides (~7KDa) with structural homology to pro-insulin. IGF2 is also a pro-survival factor, protecting against apoptosis [5–7]. Although IGF expression is ubiquitous in many cell types, they are most abundant in the cells and tissues of mesodermal origin, and form important components of stem cell niches [8–13]. *Igf2* transcription is regulated by genomic imprinting [14], an epigenetic process that causes a subset of genes in the genome to be transcribed according to parental origin. Imprinted genes regulate key aspects of mammalian physiology, from growth to energy homeostasis [15]. In most mouse and human tissues, *Igf2* is only transcribed from the

paternally inherited allele, with the maternal allele being repressed by a methylation sensitive CTCF-dependent boundary that restricts the access of downstream enhancers to the *Igf2* gene promoters [16]. In humans, reduced *Igf2* expression contributes to the intra-uterine growth restriction in patients with Silver-Russell syndrome [17]. Conversely, bialellic *Igf2* expression caused by loss of *Igf2* imprinting is observed in Beckwith-Wiedemann patients [18], a syndrome characterized by somatic overgrowth, neonatal hypoglycaemia with variable penetrance and increased predisposition to tumours. In mice, increased supply of embryonic *Igf2* above biallelic expression can result in disproportionate overgrowth associated with heart enlargement, oedema and fetal death [19]. *Igf2* null mice are viable, depending on the genetic background, but have almost half the adult body weight [19].

Key questions about how IGF2-mediated growth effects are developmentally programmed and how IGF2 contributes to organ growth control *in-vivo* remain unanswered. The normal timing, location and duration of IGF2 supply at the organ level are likely to be crucially important. The pancreas serves as an interesting model for the study of these processes, as it requires tight spatial and temporal regulation of proliferation, differentiation and morphogenesis. Very little is known about the main sites of *Igf2* expression and imprinting during pancreas development. Moreover, most mouse transgenic studies conducted so far have been directed to the beta-cells, in which *Igf2* is overexpressed [20,21] or knocked-out [22], or focused on mice that overexpress *Igf2* [23,24] or lack *Igf1* and *Igf2* constitutively [25]. Here we describe the generation of a conditional knock-out allele for *Igf2* and perform analyses of cell-type specific deletions that target the developing pancreatic mesenchyme (*Nkx3.2*-Cre), or epithelium (*Ptf1a*-Cre), with the overall aim of defining the paracrine and autocrine roles of IGF2 in pancreatic growth.

## Results

### The mesenchyme-derived cells are the main source of *Igf2* in the developing perinatal and postnatal pancreas

We first sought to establish which major cell types within the developing mouse pancreas express *Igf2* transcripts and to determine their relative levels. To achieve this aim, we used a recombinase inducible YFP reporter under the control of the *Rosa26* locus (*Rosa26YFP*-stop^fl/fl) [26] and crossed this mouse model with transgenic strains that express Cre recombinase in the main cell types of the developing pancreas, i.e. *Nkx3.2*-Cre: mesenchyme [27]; *Ptf1a*-Cre: pancreatic epithelium [28]; *RIP*-Cre: beta-cells [29]. We confirmed the activity of each Cre line by performing immunofluorescence staining for YFP in combination with cell-type specific markers in histological sections (Fig 1A). We then performed an expression timeline analysis based on fluorescence activated cell sorting (FACS) of YFP-positive cells isolated from pancreas at various developmental stages, ranging from embryonic day 16 (E16) to adulthood, followed by qRT-PCR measurements of *Igf2* mRNA levels (Fig 1B, S1A Fig). This analysis revealed that mesenchyme-derived cells express the highest levels of *Igf2* mRNA when compared to beta-cells or non-mesenchyme cells (which is comprised of acinar, endocrine, ductal and endothelial cells) (Fig 1B). At E16, mesenchyme-derived cells express 380 fold more *Igf2* than beta-cells and 1.5 fold more than non-mesenchyme cells. At P14, the difference between mesenchyme and non-mesenchyme is the greatest, with mesenchyme-derived cells expressing 700 fold more *Igf2* than non-mesenchyme cells. The levels of *Igf2* in mesenchyme-derived cells remain high in the neonatal period until the weaning period (at P21 levels are 3.2 fold reduced compared to E16, followed by a steep decline in adulthood) (Fig 1B). Interestingly, both non-mesenchyme cells and the beta-cells decrease their *Igf2* expression levels after E16. To verify that the mesenchyme-derived cells express high levels of *Igf2* mRNA, we next performed *in*

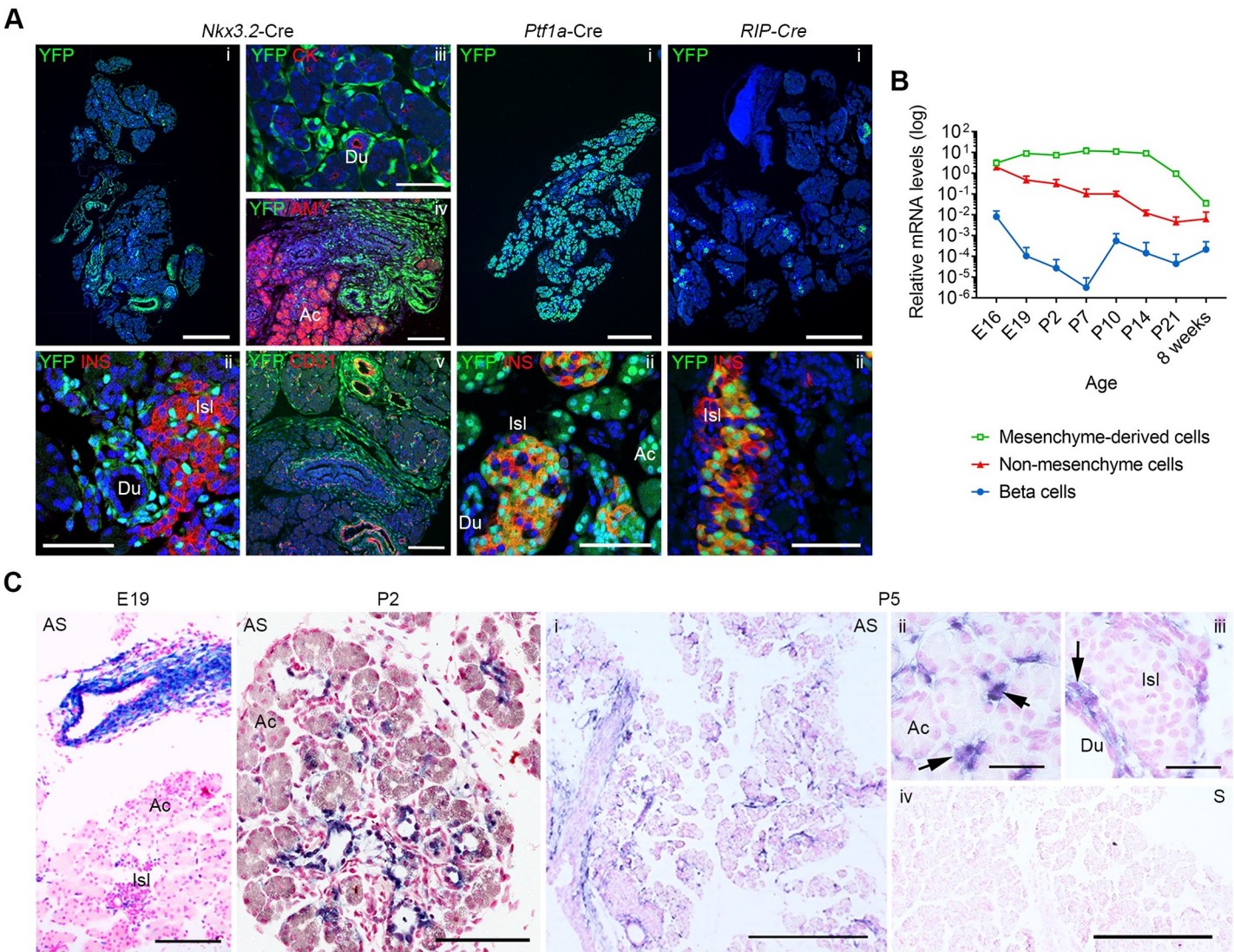

**Fig 1. Developmental profiling of cell type specific *Igf2* mRNA expression in mouse pancreas.** (A) Assessment of Cre recombinase specificity using the *Rosa26YFP*-stop[fl/fl] reporter mouse line at postnatal day 2. Cells with a functional Cre recombinase express YFP protein (green). *Nkx3.2*-Cre is active in mesenchyme-derived cells (pericytes, smooth muscle cells, stromal cells surrounding pancreatic ducts and acini, fibroblasts and myofibroblasts), *Ptf1a*-Cre in pancreatic epithelium (acinar cells, pancreatic islets and ducts) and *RIP*-Cre in pancreatic beta-cells. Sections were co-stained with insulin (INS) marker of pancreatic beta-cells; cholecystokinin (CK), marker of epithelial cells lining pancreatic ducts; amylase (AMY) marker of acinar cells and CD31, marker of endothelial cells. Scale bars are as following: 500 μm in panels i corresponding to each Cre-line; 50 μm in panels ii corresponding to each Cre-line; 100 μm (panel iii) and 200 μm (panels iv and v) for *Nkx3.2*-Cre. (B) Timeline of *Igf2* mRNA expression measured by qRT-PCR in YFP+ or YFP–FACS isolated cells from offspring of *Nkx3.2*-Cre or *RIP*-Cre females mated with *Rosa26YFP*-stop[fl/fl] males (E–embryonic day; P–postnatal day). Highest levels of *Igf2* expression are observed in the mesenchyme fraction (*Nkx3.2*-Cre YFP+ cells) throughout development. Expression data is normalized to *Ppia* and is shown as average + SD (n = 3–5 per time point and cell fraction). (C) *Igf2* mRNA analysis by *in situ* hybridization at E19, P2 and P5 showing high levels of expression in mesenchymal cells (purple colour). AS–antisense probe; S–sense probe (negative control); Ac–acinar cells; Isl–pancreatic islet; Du–pancreatic duct; arrows point to cells expressing high levels of *Igf2* mRNA. Scale bars are as follows: E19 (300 μm); P2 (100 μm); P5 (30 μm for higher resolution panels ii and iii and 300 μm for lower resolution panels i and iv).

*situ* hybridization analysis at E19, P2 and P5. Our results confirm that *Igf2* transcripts are localized in mesenchyme-derived cells surrounding the vessels, pancreatic acini and ducts, with low to undetectable expression in endocrine and acinar cells (Fig 1C).

The results presented above suggest that *Igf2* is a mesenchymal-derived cell marker in the neonatal and post-natal period, with mostly restricted expression to these cell types compared

to non-mesenchyme. To confirm this observation, we next performed RNA-seq analyses in mesenchyme-derived cells and non-mesenchymal cells isolated by FACS from pancreases of *Rosa26YFP*-stop$^{+/fl}$; *Nkx3.2*$^{Cre/+}$ mice at P2. We found over four thousand differentially expressed genes (DEGs, fold change >1.5; FDR adjusted p value <0.05) between the mesenchyme-derived cells and non-mesenchyme cells in wild-type *Igf2*$^{+/+}$ pancreases (S2A Fig, S1 Data). A number of these DEGs were validated by qRT-PCR in independent biological replicates (S2B Fig). Using DAVID functional annotation (see Materials and Methods) we identified over 20 significantly enriched GO terms (FDR adjusted p value <0.05), which included biological processes known to be involved in pancreatic mesenchyme function (e.g. Wnt signalling), early pancreas development (e.g. retinoic acid and SMAD signalling) and epithelial-mesenchymal interactions (integrin-mediated signalling, cell adhesion, collagen catabolism) (Fig 2A). This analysis also highlighted unexpected pathways, such as semaphorin-plexin and insulin-like growth factor receptor signalling (Fig 2A). Notably, several gene members of the IGF signalling family are highly enriched in mesenchyme-derived cells versus non-mesenchyme (Fig 2B). Interestingly, imprinted genes, with known roles in growth control, make up almost one third of the top 30 genes most highly expressed genes in the pancreatic mesenchyme-derived cells, with *Igf2* being the highest (Fig 2C) and one of the most enriched compared to non-mesenchyme cells (53 fold) (S2A Fig, S1 Data). DAVID functional annotation identified 17 GO terms enriched in non-mesenchyme cells including expected pathways, such as digestion and endocrine pancreas development, as well as novel pathways, such as peptidyl-tyrosine phosphorylation (S2C Fig, S1 Data). Contrary to mesenchyme-derived cells, no imprinted genes are found amongst the top 30 most highly expressed genes in non-mesenchyme cells (S2D Fig).

The observation of high levels of expression in the mesenchyme-derived cells isolated at P2 for several imprinted genes, including *Igf2*, prompted us next to extend our analysis to earlier developmental time points. We analysed the recently published transcriptomes of E13.5, E15.5 and E17.5 pancreatic mesenchyme-derived cells [3] isolated by FACS in *Rosa26YFP*-stop$^{+/fl}$; *Nkx3.2*$^{Cre/+}$ genetic crosses, as also used in our study. Similar to our observations at P2, the eight imprinted genes exhibit very high levels of expression at all three embryonic stages (Fig 2D) and *Igf2* ranks among the top 22 most highly expressed genes. Additionally, all gene members of the IGF signalling family identified in our analysis at P2 were expressed in E13.5, E15.5 and E17.5 pancreatic mesenchyme-derived cells, with ten of these genes reaching levels above 50 FPKM at all three developmental time points (Fig 2E). Our analysis further shows that at all three embryonic stages, as well as at P2, the pancreatic mesenchyme-derived cells express not only pericyte markers, as previously published [3], but also markers of other cell types, including smooth muscle cells, fibroblasts and myofibroblasts (S3 Fig), and in agreement with the highly heterogeneous histological features that we observed at P2 (Fig 1A).

To accurately measure the contribution of the various cell types to the overall pancreatic *Igf2* levels in the neonatal period, we analysed cell-type specific deletions of *Igf2* using floxed mice (*Igf2*$^{+/fl}$) that we generated (S4 Fig, S5 Fig). Deletion of *Igf2* from mesenchymal-derived cells reduced its total pancreatic levels by more than 80% (Fig 2F). Notably, mesenchyme-derived cells represent 10.3 ± 1.2% (n = 7) of the pancreatic tissue weight at this time point, as measured by stereology. *Ptf1a*-Cre mediated deletion of *Igf2* in exocrine, endocrine and ductal cells (S1B Fig) showed no discernible changes in the whole pancreas *Igf2* mRNA, which is consistent with *in situ* hybridisation data shown (Fig 1C). Deletion of *Igf2* from endothelial cells (mediated by *Tek*-Cre [30]) reduced *Igf2* mRNA with approximately 20–25% (Fig 2F). Altogether, these results demonstrate that the mesenchyme-derived cells are the main source of *Igf2* in the pancreas during late gestation and early postnatal life.

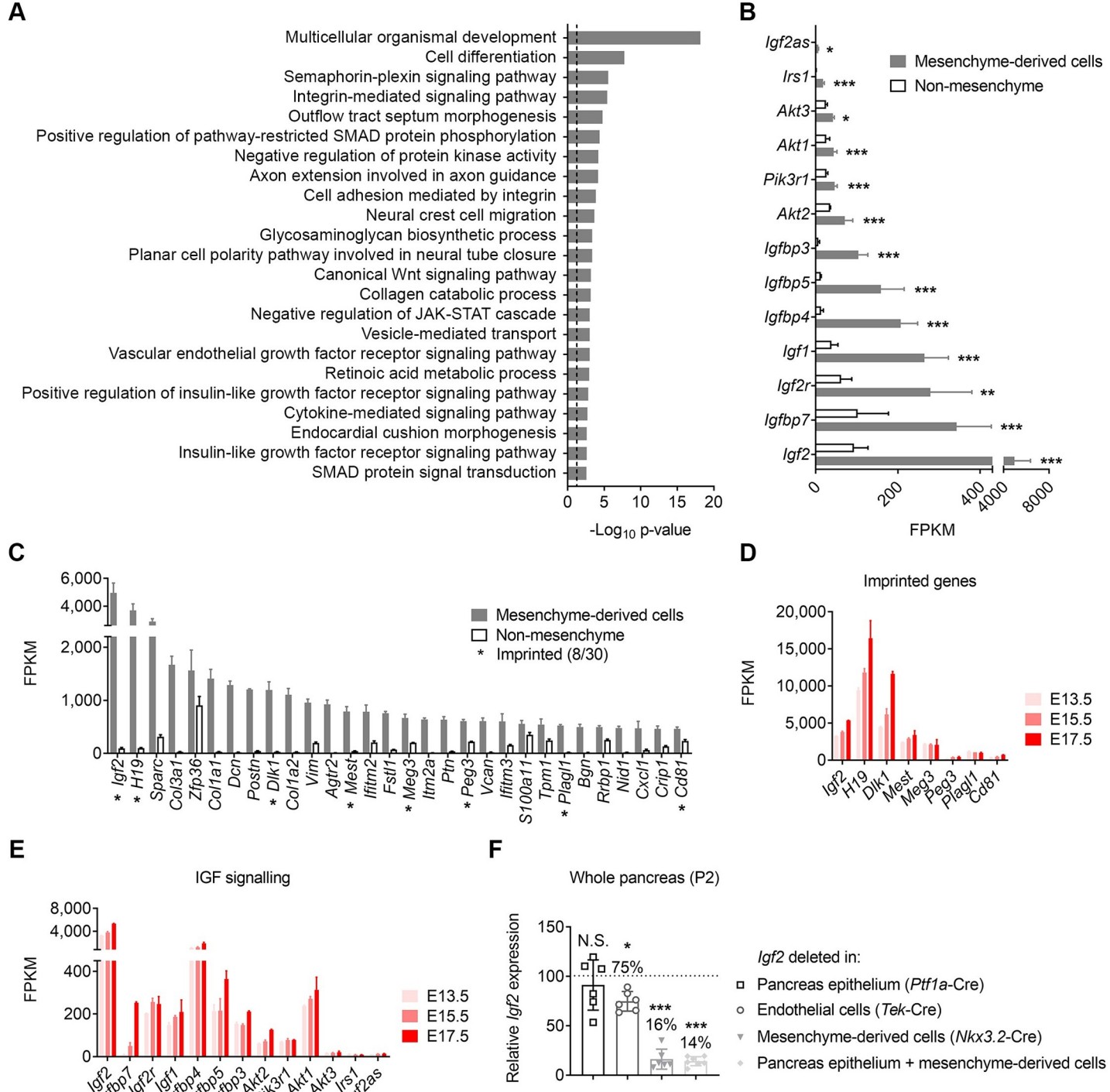

**Fig 2. Perinatal pancreatic mesenchyme-derived cells are enriched in imprinted genes (including *Igf2*) and genes related to IGF signalling.** (A) Top scoring biological processes containing genes enriched in the pancreatic mesenchyme-derived cells as identified by DAVID functional annotation. The dotted line in panel corresponds to a p value of 0.05. (B) Genes related to IGF signalling that are significantly enriched in mesenchyme-derived cells. Data is shown as average FPKM + SEM (n = 4), ranked by levels of expression. FDR adjusted p values: * p<0.05, ** p<0.01 and *** p<0.001. (C) Top 30 expressed genes with highest FPKM values in pancreatic mesenchyme-derived cells (average FPKM + SEM in mesenchyme and non-mesenchyme cells with n = 4 per group) at P2. Note that all 30 genes are significantly enriched in mesenchyme compared to non-mesenchyme (>1.5 fold, FDR adjusted p value <0.05). Asterisks indicate imprinted genes, of which there are 8 amongst the top 30. (D) RNA-seq data showing high levels of expression in mesenchyme-derived cells (as FPKM ± SEM) during embryonic life (E13.5, E15.5 and E17.5, n = 2 per developmental stage; data derived from [3]) of the top eight imprinted genes identified at P2 as shown in panel (C). (E) RNA-seq data showing high levels of expression in mesenchyme-derived cells (as FPKM ± SEM) during embryonic life (E13.5, E15.5 and E17.5, n = 2 per developmental stage; data derived from [3]) of genes related to

IGF signalling identified at P2 as shown in panel (B). (F) Residual *Igf2* mRNA levels after conditional *Igf2* deletion in different pancreatic cell types, measured by qRT-PCR in whole pancreas lysates at P2 from offspring of heterozygous Cre females mated with *Igf2*$^{+/fl}$ males. Residual levels of *Igf2* mRNA in each mutant are indicated as percentage values of controls. Data was normalized to *Ppia* and shown as individual values and averages ± SD relative to levels in control littermates, which were arbitrarily set to 100; (n = 6–7 samples per genotype; N.S.–non-significant; * p<0.05; *** p<0.001 by unpaired Student's *t*-test).

### Mesenchyme-specific *Igf2* signalling controls the growth of exocrine and endocrine pancreas

To assess whether mesenchymal *Igf2* plays a role in controlling the early growth of the pancreas, *Igf2*$^{+/fl}$*; Nkx3.2*$^{Cre/+}$ knockout mice were first analysed at the neonatal stage P2 (Fig 3A–3D). As expected, paternal transmission of the mutation resulted in 99% reduction of *Igf2* levels at the mRNA level in FACS-isolated mesenchymal cells (Fig 3A), and absence of protein by immunofluorescence (Fig 3B). Mice with deletion of the paternal *Igf2* allele have normal body weight, but significantly lighter pancreases (69% of normal) compared to littermate controls (Fig 3C, S6A Fig). Deletion of the maternal *Igf2* allele has no phenotypic effects, which is consistent with findings that the maternal allele is transcriptionally inactive due to imprinting (S6B and S6C Fig). Importantly, *Nkx3.2*-Cre or *Igf2* floxed carrier mice are indistinguishable from wild-type littermates (S6A and S6B Fig). We next measured the number of cell nuclei in

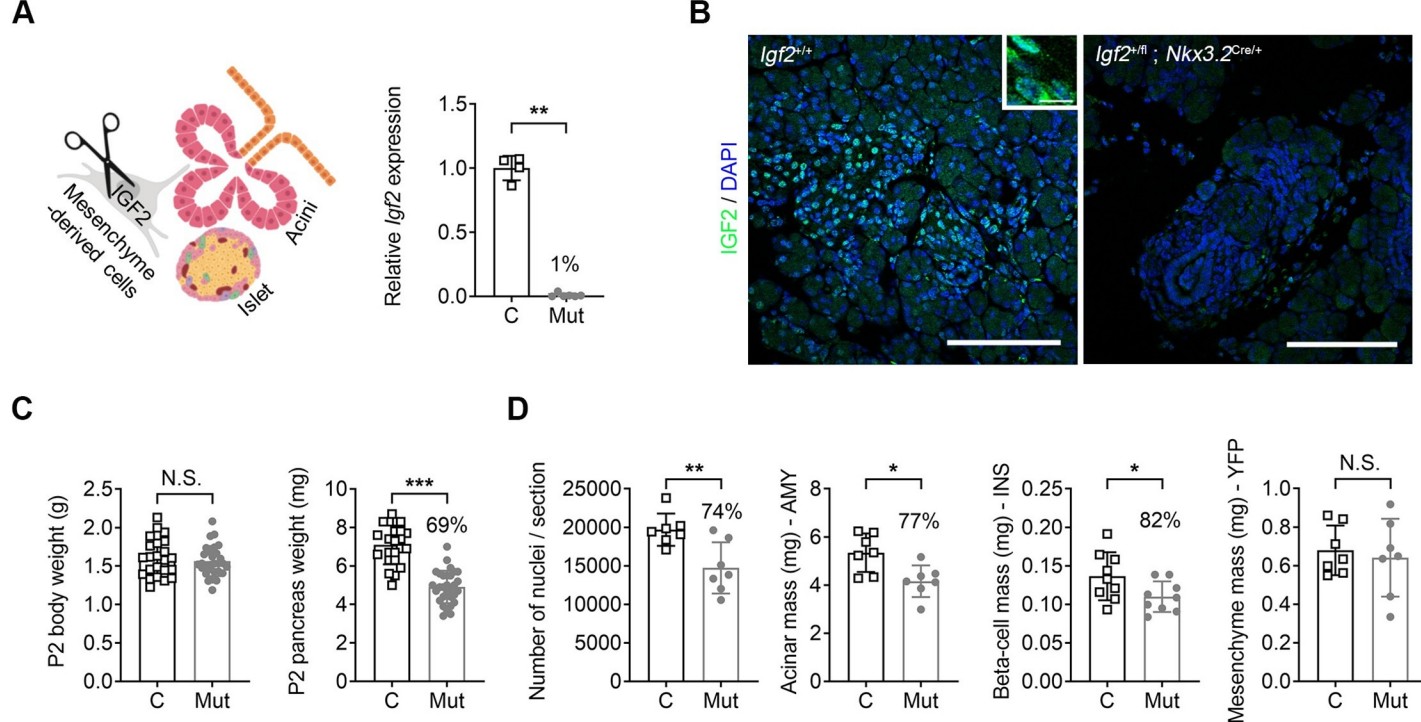

**Fig 3. Impact of mesenchyme-specific deletion of *Igf2* on pancreas at postnatal day 2 (P2).** (A) Left: schematic representation of conditional *Igf2* deletion from the pancreatic mesenchyme. Right: reduced *Igf2* mRNA expression levels in FACS-isolated pancreatic mesenchyme-derived cells of mutant mice (Mut: *Igf2*$^{+/fl}$; *Rosa26YFP*-stop$^{+/fl}$; *Nkx3.2*$^{Cre/+}$) compared to control littermates (C: *Igf2*$^{+/+}$; *Rosa26YFP*-stop$^{+/fl}$; *Nkx3.2*$^{Cre/+}$) measured by qRT-PCR (data normalized to *Ppia* and shown relative to levels in controls). (B) Representative immunofluorescence confocal microscopy showing cytoplasmatic staining (inset) for IGF2 in pancreases of control mice (*Igf2*$^{+/+}$) and absence of signal in mutants (*Igf2*$^{+/fl}$; *Nkx3.2*$^{Cre/+}$). Green–IGF2; blue–DAPI staining of nuclei; scale bars– 100 μm and 10 μm for lower and higher magnification panels, respectively. (C) Pancreas weights are significantly reduced in mutants with conditional deletion of *Igf2* in the mesenchyme (Mut: *Igf2*$^{+/fl}$; *Rosa26YFP*-stop$^{+/fl}$; *Nkx3.2*$^{Cre/+}$), compared to littermate controls (C: *Igf2*$^{+/+}$; *Rosa26YFP*-stop$^{+/fl}$). Body weights are similar between genotypes. (D) Stereological measurements show significant reductions in the total number of nuclei/section, acinar mass and beta-cell mass (shown as %) in mutants (Mut: *Igf2*$^{+/fl}$; *Rosa26YFP*-stop$^{+/fl}$; *Nkx3.2*$^{Cre/+}$), compared to littermate controls (C: *Igf2*$^{+/+}$; *Rosa26YFP*-stop$^{+/fl}$; *Nkx3.2*$^{Cre/+}$), but not for the mesenchyme mass. For all graphs, data is shown as individual values and averages ± SD; N.S.–non-significant, * p<0.05, ** p<0.01 and *** p<0.001 by unpaired Student's *t* tests [panels (A), (C), and (D)].

DAPI-stained paraffin sections to establish if the loss of pancreas weight was due to hypoplasia. We found that the number of cell nuclei was significantly reduced in mutants (74% of normal) compared to controls (Fig 3D). This finding suggests that mutant pancreases are smaller due a reduction in cell numbers, which is in agreement with the well-established role for IGF2 in the control of cell proliferation and survival. Stereological analysis revealed that loss of *Igf2* in pancreatic mesenchyme leads to decreases in acinar and beta-cell mass, in line with the overall pancreas weight deficit, while the mesenchymal mass remained unchanged (Fig 3D). Also consistent with the decrease in acinar mass, mutant pancreases contained lower amounts of lipase (11,367 ± 5674 in n = 6 mutants versus 18,343 ± 4238 in n = 7 controls, p = 0.035). Thus, our data reveals that the mesenchyme-derived IGF2 is likely to control acinar and beta-cell growth through paracrine signalling from the mesenchyme.

The microRNA *miR-483* is located within the intron 4 of *Igf2*, which is deleted alongside the *Igf2* coding exons 4–6 upon Cre-mediated recombination. To rule out a contributory role of this microRNA to the phenotype, we analysed pancreases of mice carrying a *miR-483* deletion [31], and found that it does not alter pancreas weight at P2 (S7 Fig). Furthermore, *mir-483* knockout mice are normal sized, viable and show no evidence of impaired glucose homeostasis defects or other discernible phenotypes up to at least three months of age (Sekita *et al.* manuscript in preparation). We therefore conclude that the growth phenotype in the *Igf2*[+/fl]; *Nkx3.2*[Cre/+] knockout mice can be attributed solely to IGF2 signalling actions.

Next, we used a well-established mouse model of *Igf2* loss-of-imprinting–the *H19*DMD conditional knockout [32]–to overexpress *Igf2* specifically in the pancreatic mesenchyme (*H19*DMD[fl/+]; *Nkx3.2*[+/Cre]) and investigated the effects on acinar growth (Fig 4A–4D). As expected, deletion of the imprinting control region (DMD) from the maternal allele resulted in increased *Igf2* and decreased *H19* RNA levels (160% and 18% of controls, respectively–Fig 4A) and increased IGF2 protein (Fig 4B), and was associated with a 29% increase in pancreas weight, with similar body weights between genotypes at P2 (Fig 4C). The acinar cell mass was increased by 34%, in line with the overall pancreas weight increase (Fig 4D). Beta-cell mass remained unchanged, while the mesenchymal mass increased by 45% (Fig 4D). We then used an *Igf2* gain-of-function model that does not lead to transcriptional changes in *Igf2* mRNA but instead results in increased IGF2 protein (Fig 4E–4G). Accordingly, we deleted the IGF-type II receptor that is required for recycling IGF2 via lysosome degradation [33], specifically in the pancreatic mesenchyme (*Igf2r*[fl/+]; *Nkx3.2*[+/Cre]) (Fig 4E). Reduced *Igf2r* mRNA levels in the mesenchyme cells (25% of controls), was associated with a 20% increase in pancreas weight, with similar body weights between genotypes at P2 (Fig 4F). The acinar cell mass was also increased by 33% in this model (Fig 4G). Additionally, beta-cell mass was also increased, but the observed difference between groups (31%) did not reach statistical significance (p = 0.099), while mesenchymal mass remained unchanged (Fig 4G). Therefore, the two gain-of-function genetic models further demonstrate that IGF2 produced by the pancreatic mesenchyme-derived cells controls the growth of the acinar cells in a paracrine manner.

## Mesenchyme-derived IGF2 exerts paracrine effects on pancreatic epithelium *in vivo* and acinar cells respond to exogenous IGF2 *in vitro*

To investigate the molecular signatures associated with the paracrine effects exerted by the mesenchyme-derived IGF2, we performed genome wide-transcriptional profiling by RNA-seq in both fractions of *Igf2*[+/fl]; *Nkx3.2*[Cre/+] mutants (*i.e.* mesenchyme-derived cells with *Igf2* deleted and the corresponding non-mesenchyme fraction), compared to mesenchyme and non-mesenchyme cells isolated from controls, at the neonatal P2 stage. We found that transcriptional changes are far more widespread in the non-mesenchyme fraction (498 DEGs with

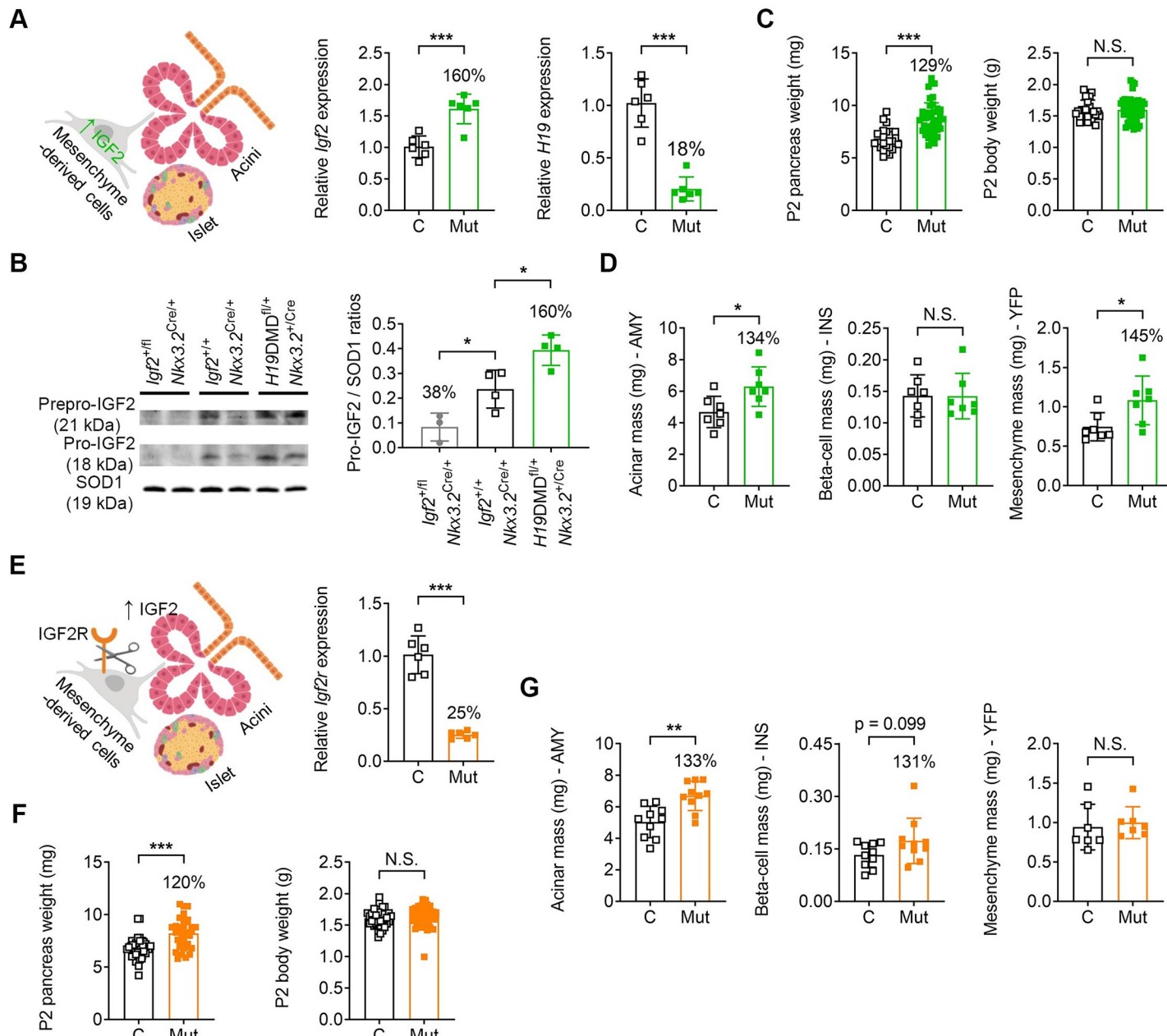

**Fig 4. Impact of mesenchyme-specific *Igf2* gain-of-function on pancreas at postnatal day 2 (P2).** (A) Left: schematic representation of conditional *Igf2* overexpression in pancreatic mesenchyme. Right: increased *Igf2* mRNA expression and reduced *H19* RNA levels in whole pancreases of mutant mice (Mut: *H19*DMD^fl/+; *Rosa26YFP*-stop^fl/+; *Nkx3.2*^+/Cre) compared to control littermates (C: *H19*DMD^+/+; *Rosa26YFP*-stop^fl/+) measured by qRT-PCR (normalized to *Ppia* and shown relative to levels in controls). (B) Left: western blot analysis of Prepro-IGF2 and Pro-IGF2 in the two models of *Igf2* loss-of-function and *Igf2* gain-of-function. Right: quantification graph for Pro-IGF2/SOD1 ratios only. (C) Pancreas weights are significantly increased in mutants with conditional *Igf2* overexpression in the mesenchyme (Mut: *H19*DMD^fl/+; *Rosa26YFP*-stop^fl/+; *Nkx3.2*^+/Cre), compared to littermate controls (C: *H19*DMD^+/+; *Rosa26YFP*-stop^fl/+). Body weights are similar between genotypes. (D) Stereological measurements show a significant increase in acinar mass and mesenchyme mass, but not in beta-cell mass in mutants with conditional *Igf2* overexpression in the mesenchyme (Mut: *H19*DMD^fl/+; *Rosa26YFP*-stop^fl/+; *Nkx3.2*^+/Cre), compared to littermate controls (C: *H19*DMD^+/+; *Rosa26YFP*-stop^fl/+; *Nkx3.2*^+/Cre). (E) Left: schematic representation of conditional *Igf2r* deletion in pancreatic mesenchyme. Right: reduced *Igf2r* mRNA levels in whole pancreases of mutant mice (Mut: *Igf2r*^fl/+; *Rosa26YFP*-stop^fl/+; *Nkx3.2*^+/Cre) compared to control littermates (C: *Igf2r*^+/+; *Rosa26YFP*-stop^fl/+) measured by qRT-PCR. Data was normalized to *Ppia* and is shown relative to levels in controls (arbitrarily set to 1). (F) Pancreas weights are significantly increased in mutants with conditional *Igf2r* deletion in the mesenchyme (Mut: *Igf2r*^fl/+; *Rosa26YFP*-stop^fl/+; *Nkx3.2*^+/Cre), compared to littermate controls (C: *Igf2r*^+/+; *Rosa26YFP*-stop^fl/+). Body weights are similar between genotypes. (G) Stereological measurements show a significant increase in acinar mass and a trend for increased beta-cell mass (P = 0.099), but similar mesenchymal mass in mutants with conditional *Igf2r* deletion in the mesenchyme (C: *Igf2r*^+/+; *Rosa26YFP*-stop^fl/+; Mut: *Igf2r*^fl/+; *Rosa26YFP*-stop^fl/+; *Nkx3.2*^+/Cre). For all graphs, data is shown as individual values and averages ± SD; N.S.–non-significant, * p<0.05, ** p<0.01 and *** p<0.001 by unpaired Student's *t* tests [panels (A), (C), (D), (E), (F) and (G)] or 1-way ANOVA followed by Dunnett's post hoc test for multiple comparisons against the *Igf2*^+/+; *Nkx3.2*^Cre/+ controls [panel (B)].

>1.5-fold change and FDR-adjusted p value <0.05) (Fig 5A, S2 Data) than in *Igf2*-deficient mesenchyme-derived cells (151 DEGs; Fig 5A, S3 Data). DAVID functional annotation identified 15 enriched GO terms in the non-mesenchyme fraction (Fig 5B, S2 Data) but only 2 in the *Igf2*-deficient mesenchyme fraction (namely, actin binding and inflammatory response, S3 Data). The top biological processes in the non-mesenchyme fraction enriched in genes up-regulated in mutants were related to digestion and inflammation/immune responses, and biological processes enriched in genes down-regulated in mutants were related to erythrocyte development and tRNA methylation (Fig 5B and 5C, S2 Data). These findings are in agreement with the stereological analyses showing that loss of *Igf2* from the mesenchyme-derived cells impacts on the growth/function of the non-mesenchymal fraction through a paracrine effect.

To provide additional evidence that mesenchyme-derived IGF2 acts on acinar cells in a paracrine manner, we next isolated primary mesenchyme-derived and acinar cells from P2 pancreases (see Materials and Methods) (Fig 5D–5F). Mesenchymal-derived cells secreted higher levels of IGF2 into the conditioned media compared to acinar cells (Fig 5G), a finding that is consistent with higher levels of *Igf2* mRNA expression in the mesenchyme-derived cells *in-vitro* (Fig 5F) and *in-vivo* (Fig 1, Fig 2). We then tested the effect of exogenous IGF2 treatment on acinar cells cultured *ex-vivo*. Isolated acinar cells treated with recombinant mouse IGF2 showed increasing levels of AKT phosphorylation (S473) in a concentration-dependent manner (Fig 5H). Additionally, treatment of acinar cells with 50 ng/ml IGF2 increased their cell numbers and led to a modest, but significant increase in amylase production (Fig 5I). Altogether, the *ex-vivo* data shows that mesenchyme-derived cells are capable of secreting IGF2 and that exogenous IGF2 induces intracellular signalling in acinar cells associated with increased enzymatic output.

## Post-weaning growth and glucose homeostasis analyses in *Igf2* pancreas-cell type specific knockouts

To investigate the impact of *Igf2* loss-of-function from the developing mesenchyme and/or epithelium on post-weaning growth and glucose homeostasis, single and double *Nkx3.2*-Cre and *Ptf1a*-Cre *Igf2* knockouts were analysed (Fig 6A). The pancreas weight deficit observed in single *Igf2*$^{+/fl}$; *Nkx3.2*$^{Cre/+}$ knockouts at P2 is maintained at weaning (S8A Fig) and is also observed at 9 weeks of age in single *Igf2*$^{+/fl}$; *Nkx3.2*$^{Cre/+}$ knockout males and females (Fig 6B). However, in contrast to the P2 time point, body weights are now reduced, by approximately 12% at P21 (S8B Fig), and by 5% and 10% for females and males, respectively, at 9 weeks of age (Fig 6C). After normalization for body weight, the mutant pancreases remain disproportionately smaller at both P21 and 9 weeks time points (S8A Fig, Fig 6B). Double *Igf2*$^{+/fl}$; *Nkx3.2*$^{Cre/+}$; *Ptf1a*$^{Cre/+}$ knockouts show similar pancreatic weights and body weight reductions to the single *Igf2*$^{+/fl}$; *Nkx3.2*$^{Cre/+}$ knockout (Fig 6B and 6C), thus suggesting that IGF2 produced by the endocrine and exocrine pancreas does not play major autocrine or paracrine growth roles that alter pancreas size. Consistent with this hypothesis, single *Igf2*$^{+/fl}$; *Ptf1a*$^{Cre/+}$ knockouts have normal pancreas sizes and body weights (Fig 6B and 6C).

Glucose homeostasis, assessed by oral glucose tolerance tests (OGTT), was unaltered in single *Igf2*$^{+/fl}$; *Nkx3.2*$^{Cre/+}$ and *Igf2*$^{+/fl}$; *Ptf1a*$^{Cre/+}$ knockouts, as well as in double *Igf2*$^{+/fl}$; *Nkx3.2*$^{Cre/+}$; *Ptf1a*$^{Cre/+}$ knockouts compared to *Igf2*$^{+/fl}$ controls at 8 weeks of age in both sexes (Fig 6D and 6E). We next assessed glucose homeostasis during pregnancy, a naturally occurring metabolic stress state. We found that 8 week old *Igf2*$^{+/fl}$; *Nkx3.2*$^{Cre/+}$ knockout females at the E15 stage of pregnancy had similar levels of glycaemia and insulinemia compared to age-matched pregnant *Igf2*$^{+/fl}$ littermate controls after fasting and prior to OGTT (time 0) (Fig 6F).

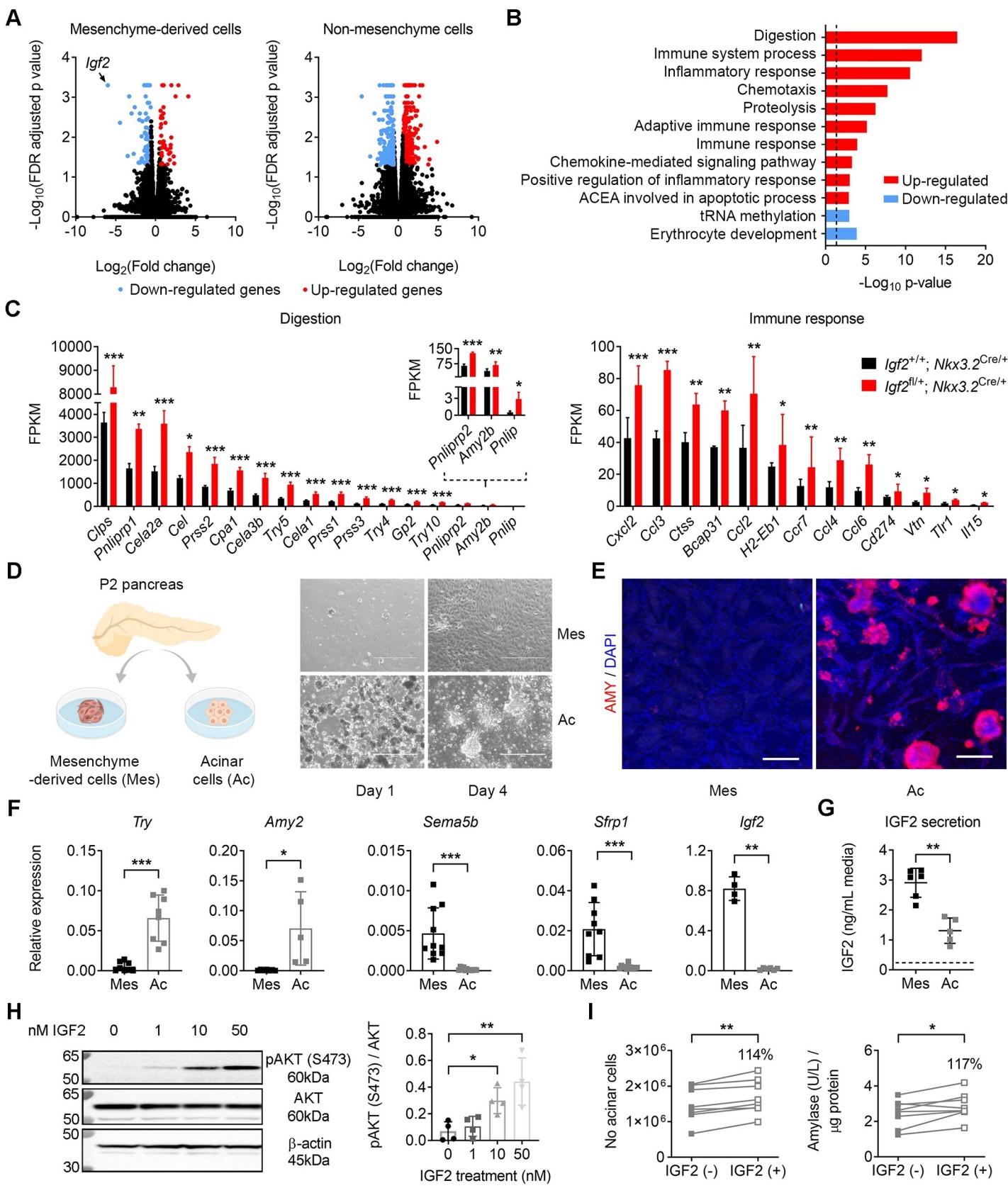

**Fig 5. Loss of *Igf2* in pancreatic mesenchyme causes significant transcriptional changes in non-mesenchymal cells *in vivo* and acinar cells respond functionally to exogenous IGF2 *ex vivo*.** (A) Volcano plot representing all expressed transcripts (by RNA-seq) in *Igf2*-deficient pancreatic mesenchyme (left) and in corresponding non-mesenchyme cell fraction (right), compared to mesenchyme and non-mesenchyme controls, respectively. Expression changes detected in the right panel thus reflect changes caused by lack of *Igf2* from the mesenchyme on neighbouring non-mesenchymal cells. Blue and red dots depict statistically significant down-regulated and up-regulated genes (fold change >1.5 fold FDR adjusted p value <0.05), respectively, and black dots show genes that are not statistically different; (n = 3 mutants and n = 4 controls per cellular fraction). There are more differentially expressed genes (DEGs) in the non-mesenchyme fraction (216 down-regulated and 282 up-regulated genes) than in the mesenchyme (109 down-regulated and 42 up-regulated genes) (all genes are listed in S2 and S3 Data). (B) Biological processes enriched in down-regulated (blue) and upregulated (red) genes between non-mesenchyme of *Igf2*$^{+/fl}$; *Nkx3.2*$^{Cre/+}$ mutants and *Igf2*$^{+/+}$; *Nkx3.2*$^{Cre/+}$ controls as identified by DAVID (for a complete list of biological processes, see S2 Data). ACEA–activation of cysteine-type endopeptidase activity. The horizontal axis shows -Log$_{10}$ of p value; the dotted line corresponds to a FDR-corrected p value of 0.05. (C) DEGs in non-mesenchymal mutant cells compared to control cells, grouped according to selected enriched biological processes shown in (B). Data is shown as average FPKM + SEM, ranked by levels of expression (the inset depicts lower expressed genes related to "digestion", with a different scale bar). FDR adjusted p values: * p<0.05, ** p<0.01 and *** p<0.001. (D) Primary mesenchyme-derived cells (Mes) and acinar cells (Ac) isolated from P2 pancreases. Representative images are shown after one and four days in culture. (E) Staining for the acinar marker amylase (AMY, red) in primary mesenchyme-derived (Mes) and acinar (Ac) cells after four days in culture. Blue–DAPI, staining of nuclei; scale bars– 100 μm. (F) mRNA expression levels of acinar (*Try*, *Amy2*) and mesenchymal (*Sema5b*, *Sfrp1*, *Igf2*) signature genes measured by qRT-PCR after four days of culture, showing high purity of the primary cell cultures. (G) Measurement of IGF2 protein secreted by mesenchymal or acinar cells in the culture media. Dotted line corresponds to background readings in media only. (H) Levels of AKT phosphorylation (pAKT-S473) after 10 minutes treatment of freshly isolated acinar cells with increasing doses of IGF2 (0nM to 50nM) by western blotting, and quantified against total AKT levels (n = 4 independent biological replicates). (I) Effect of IGF2 treatment [IGF2 (+): 50 ng IGF2/ml] of primary acinar cells on total cell number (left panel) compared to non-treated cells [(IGF2 (-)], and amylase production (U/L) normalized per protein content (right panel). For all graphs shown in panels (F)–(I), data is shown as individual values and/or averages, error bars represent SD. * p<0.05, ** p<0.01, *** p<0.001 by Mann-Whitney tests [(F), (G)], Friedman's test with Dunn's correction for multiple comparisons (H) and the Wilcoxon matched-pairs signed rank test (I).

However, during OGTT, pregnant *Igf2*$^{+/fl}$; *Nkx3.2*$^{Cre/+}$ knockout females became glucose intolerant compared to pregnant *Igf2*$^{+/fl}$ littermate controls (Fig 6F). Beta-cell mass, measured at E15, was significantly smaller in pregnant *Igf2*$^{+/fl}$; *Nkx3.2*$^{Cre/+}$ knockout females compared to pregnant *Igf2*$^{+/fl}$ littermate controls (Fig 6G). Thus, our results show that mesenchyme-specific *Igf2* deletion results in postnatal whole-body growth restriction and maternal glucose intolerance during pregnancy.

## Discussion

This study uncovers novel roles for IGF signalling in the developmental regulation of adult pancreas size, with consequences for energy homeostasis and post-natal whole body-growth. We first observed that gene members of the IGF signalling pathway, which include IGF2 and IGF1 ligands, are highly enriched in the neonatal mesenchyme-derived cells compared to the non-mesenchyme fraction. Moreover, IGF2 is the top expressed gene in the mesenchyme-derived cells, as revealed by RNA-seq profiling at postnatal day 2 (P2) and one of the top ~25 most highly expressed genes at E13.5, E15.5 and E17.5. Mesenchyme-derived cells express the highest levels of *Igf2* mRNA compared to all other pancreatic cell types from E16 to 8 weeks postnatally. At P2 we estimate, based on conditional deletions of *Igf2* from the diverse cell types, that ~75–85% of total *Igf2* in the pancreas is expressed by the mesenchyme-derived cells, ~20–25% by the endothelium and less than 1% is derived from the epithelium. Therefore, we hypothesized that mesenchymal IGF2/IGF signalling may play previously unanticipated roles in pancreatic development and generated conditional specific mouse models mainly targeting IGF2 levels in the mesenchyme.

We found that mesenchymal IGF2 is principally required for normal growth of the exocrine pancreas. Deletion of *Igf2* from the mesenchyme cells from E9.5 leads to hypoplasia of the postnatal pancreas (~69%N at P2, ~67%N at P21 and ~79–83%N at 9 weeks of age). The acinar mass is affected the most and is reduced, in line with the overall loss in pancreatic weight. This finding is in agreement with the reports of reduced exocrine mass in mice that lack both IGF ligands (*i.e.* mice lacking both *Igf1* and *Igf2* in all cells of the body), or their receptors (*i.e.* mice constitutively lacking both *Insr* and *Igf1r*) [25]. However, the contribution of the individual ligands or receptors to the loss of exocrine mass has remained unclear, mainly because *Insr* or *Igf1r* single total knockouts result in either normal or increased exocrine mass, respectively,

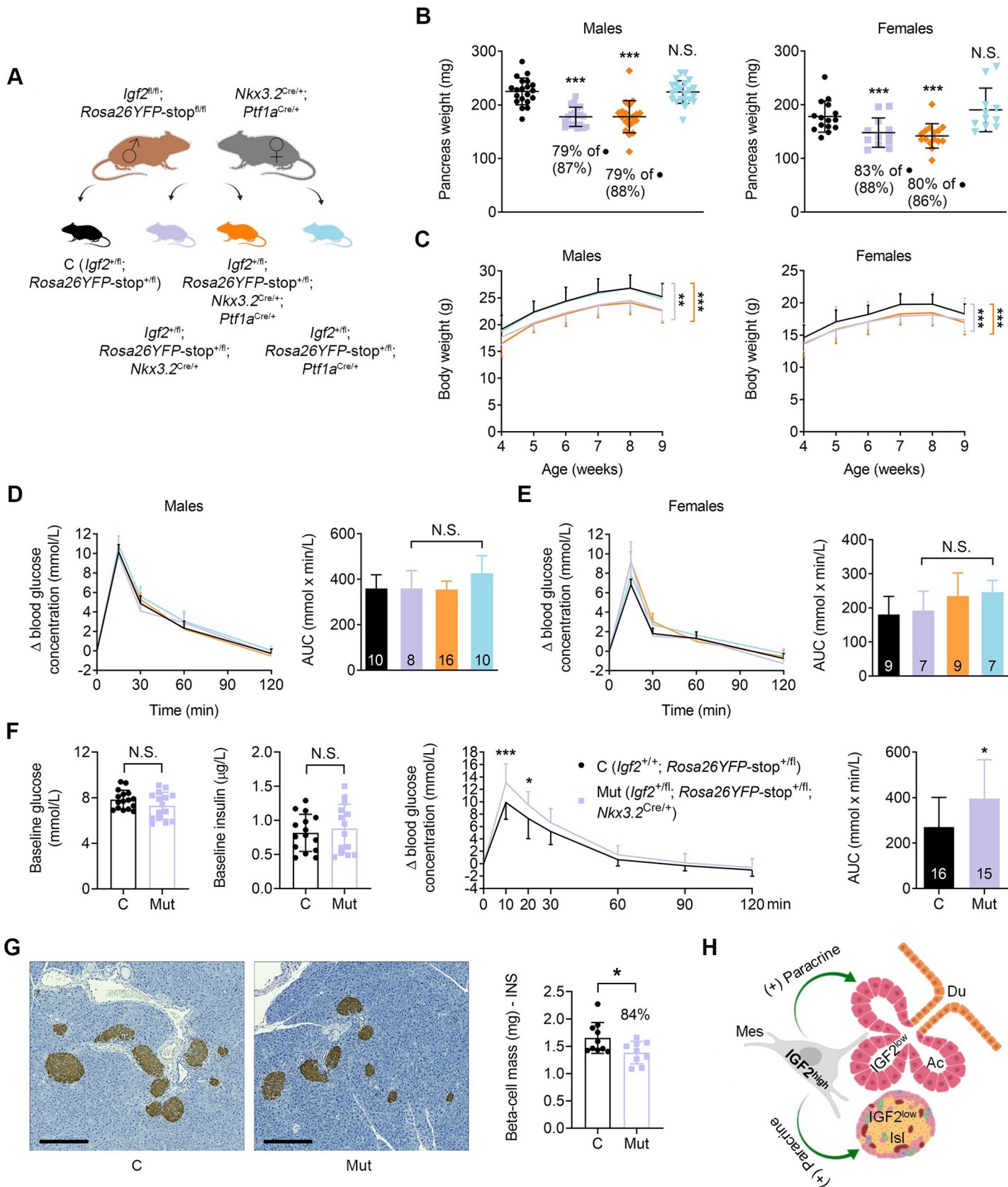

**Fig 6. Growth and glucose homeostasis regulation in pancreas-cell type specific *Igf2* knockout mice in adulthood.** (A) Schematic representation of the mating strategy used to generate control mice (black) and littermates with *Igf2* deletion in the pancreatic mesenchyme only (purple), mesenchyme plus epithelium (orange) or epithelium only (blue). (B) Pancreas weights at 9 weeks of age for males (n = 20–22 per genotype) and females (n = 11–15 per genotype). Significant reductions in weight (shown as %) are only observed in mice that carry a deletion of *Igf2* in the mesenchyme; values within brackets show % reduction after normalization to body weight; *** p<0.001 by 1-way ANOVA with Dunnett's multiple comparisons tests against controls. (C) Growth kinetics from weaning to 9 weeks of age for males (n = 20–22 per genotype) and females (n = 11–15 per genotype). ** p<0.01 and *** p<0.001 by repeated measures 1-way ANOVA with Dunnett's post hoc test for multiple comparisons against the controls. Oral glucose tolerance tests performed in 8-week old males (D) and females (E) fed chow diet, after six hours of fasting. (F) First two graphs on left side: baseline glucose and insulin levels measured after six hours fasting (at T0), before the start of the oral glucose tolerance test (OGTT) (n = 16 *Igf2*$^{+/fl}$ controls and n = 15 *Igf2*$^{+/fl}$; *Nkx3.2*$^{Cre/+}$ mutants). Third graph: OGTTs performed in 8-week old *Igf2* mesenchyme-specific deficient pregnant females (E15 of gestation). Fourth graph: area under curve (AUC). (G) Left: insulin staining (brown) and hematoxylin counterstaining (blue) in sections collected from E15 pregnant females (scale bars are 250 μm). Right: beta-cell mass quantification (n = 10 *Igf2*$^{+/fl}$ controls and n = 9 *Igf2*$^{+/fl}$; *Nkx3.2*$^{Cre/+}$ mutants). For (D), (E) and (F, third graph) changes in blood glucose concentrations (y-axis) from basal pre-treatment values with time (x-axis) after glucose administration are shown. The graphs on the right side indicate AUC, calculated using the trapezoid rule. For all panels, data is shown as individual values or average values ± SD; N.S.–non-significant. * p<0.05 and *** p<0.001 by unpaired Student's *t* tests [panel (G) right side and panel (H) right side], or two-way ANOVA with Sidak's multiple comparison tests [panels (B), (C), (D), (E) and (G) left side]. (H) Suggested model of mesenchymal IGF2 actions in the developing pancreas: IGF2 expression is high (IGF2$^{high}$) in the mesenchymal cells (Mes) of the pancreas and low (IGF2$^{low}$) in other pancreatic cell types (Ac–acinar cells, Isl–pancreatic islet cells, Du–pancreatic duct cells). IGF2 produced by the mesenchymal cells exerts paracrine effects (green arrows) on pancreatic acinar cells and pancreatic islets.

and single *Igf1* or *Igf2* total knockouts were not studied for pancreatic phenotypes. Our work now demonstrates that IGF2 is a key factor promoting acinar growth during development. Crucially, we have ruled out a role for IGF2 expressed in the developing epithelium, which includes IGF2 expressed in acinar cells, as a determinant of exocrine pancreas growth. Accordingly, deletion of *Igf2* using *Ptf1a-Cre*, results in normal pancreatic growth. We therefore conclude that the effect of IGF2 on pancreas growth is restricted primarily to signalling from the mesenchyme.

We next addressed the question of how mesenchymal IGF2 regulates pancreas size. We propose that this is mainly achieved through secretion of IGF2 from the mesenchyme and signalling to the neighbouring cells types in a paracrine manner (Fig 6H). This hypothesis is based on the following findings: in primary cell cultures we show that mesenchyme-derived cells secrete IGF2 into the culture media, and that exogenous IGF2 stimulates activation of a key signalling node, AKT, downstream of IGF1R and INSR in acinar cells, increasing their number and enzymatic production. *In vivo*, increased IGF2 levels specifically in the mesenchyme led to significant overgrowth of the exocrine pancreas, as shown in two distinct genetic models (*H19*DMD$^{fl/+}$; *Nkx3.2*$^{+/Cre}$, and *Igf2r*$^{fl/+}$; *Nkx3.2*$^{+/Cre}$). Importantly, deletion of *Igf2* from the mesenchyme leads to a reduction in beta-cell mass (~82% N at P2 and ~84% N in pregnant females at E15), showing that the paracrine effects are not exclusive to the exocrine pancreas. We found that mesenchyme mass at postnatal day 2 was not affected by the deletion of mesenchymal *Igf2* or overexpression of IGF2 in the mesenchymal *Igf2r* mutant model. These findings strongly support the hypothesis that the effects of the mesenchyme are mainly paracrine, thus suggesting that a) the pancreas growth restriction occurs independently from the loss of mesenchyme-derived cells; b) the function of the IGF2 producing mesenchyme-derived cells is mainly to act as a developmental reservoir for IGF2 paracrine signalling. The lack of mesenchymal IGF2 has important physiological consequences. It is well established that the pancreas size must match the physiological demands of the host organism to promote nutrient digestion and absorption in the gut and to maintain glucose homeostasis. In this study, we report that mesenchyme *Igf2*-deficient mice have a disproportionately smaller pancreas, with a 30% size reduction at P2, which is associated with a reduced output of secreted acinar enzymes, such as lipase. Our data suggests that nutrient malabsorption due to loss of exocrine mass and/or exocrine dysfunction might be causative of the postnatal growth restriction, which is first observed in these mice at weaning and maintained throughout adulthood. Mouse models with perturbations of lipase genes such as colipase (*Clps*) and pancreatic-lipase related protein 2 (*Pnliprp2*) exhibit a similar body weight growth defect phenotype due to an

inability to process fat from mother's milk [34,35]. It has been suggested that the lipase defi-ciency can programme a "set-point" of body weight, with an inability to catch-up in body weight later, due to the effects of poor weight gain during the suckling period [34,35]. Interest-ingly, RNA-seq shows that mesenchyme *Igf2*-deficient pancreases upregulate transcription of exocrine genes involved in the production of digestive enzymes. We suggest that these changes represent adaptive responses of the small exocrine pancreas to meet the demand for digestion and absorption of nutrients in early postnatal life. However, this compensatory mechanism might not be sufficient to match the demand, and whole-body growth restriction ensues. The RNA-seq studies also showed increased expression of chemokines, which are normally up-reg-ulated by pro-inflammatory stimuli and attract immune cells by inducing a response from the innate and adaptive immune system [36,37]. Whether a pre-pancreatitis state exists, linked to the premature activation of digestive enzymes in the interstitial space of the pancreas, and how such a state might contribute to putative nutrient absorption abnormalities require further investigation. We also observed that the loss of IGF2 paracrine effects from the mesenchyme results in a modest reduction of the beta-cell mass that does not impact on glucose homeostasis control in young mice under a normal diet and physiological state. However, induced meta-bolic stress, as it occurs naturally during pregnancy, leads to glucose intolerance. We suggest that the loss of beta-cell mass in the mutant pregnant females compared to controls is a con-tributing factor for the observed glucose intolerance. Further work is required to understand the precise mechanisms, which includes the generation of a double knockout of *Igf2* from the mesenchyme and the beta-cell. It is possible that a more severe glucose homeostasis in preg-nancy will occur when the known autocrine IGF2 actions in the beta-cell are removed [22].

Our findings offer novel insights into mesenchymal factors that regulate pancreas organo-genesis. Wnt, BMP, FGF and Hedgehog signalling pathways have been implicated as early mesenchymal factors regulating pancreas development and growth [2,38–42], but for the majority of these studies the specific roles in the mesenchyme could not be assessed (it has only recently become possible to start interrogating the roles that genes play in the mesen-chyme with the generation of specific Cre recombinases). To our knowledge, this study, using a combination of loss and gain-of-function conditional mouse models, is the first to identify a mesenchymal-specific paracrine growth signal of the developing pancreas. It also reveals that IGF signalling plays previously unappreciated roles in the mesenchyme function and in the epithelial-mesenchymal interactions during pancreas organogenesis. *Igf1*, like *Igf2*, is highly enriched in the mesenchyme-derived cells compared to non-mesenchyme cells and paracrine release of IGF1 from fibroblasts, which derive from the primitive mesenchyme, stimulate aci-nar cell proliferation during regeneration from acute pancreatitis [43]. Interestingly, deletion of *Igf1* from the developing epithelium also results in normal exocrine pancreas growth [44], similarly to the results reported here for *Igf2*. Our study has also highlighted a number of growth-related imprinted genes (including *Plagl1/Zac1* a master regulator of an imprinting growth network [45]) that are highly enriched in the mesenchyme, and raises the interesting hypothesis that genomic imprinting contributes significantly to mesenchymal-specific func-tions during pancreas development.

Our postnatal studies provide proof of principle that lack of mesenchyme-derived IGF2 has physiological consequences related to whole-body growth (first observable around weaning) and glucose intolerance during pregnancy. It remains to be established if pregnancy-related beta-cell mass expansion is impaired and the extent to which other metabolic stress states, such as ageing, high fat diet, acute insulin resistance states, may uncover mesenchymal IGF2 effects on beta-cell function.

Our study has a number of limitations. Although we provide evidence that mesenchymal IGF2 is a long-sought promotor of acinar growth, we have not established the precise timing

of the actions of IGF2. IGF2 is well known to promote cell proliferation and survival, with the main wave of these actions thought to occur in the mouse between E9-E10 [46]. Therefore, it is likely that many of the mesenchymal IGF2 actions in promoting pancreatic growth start during the earliest stages of pancreas development. However, based on RNA-seq data we also observed very high levels of *Igf2* expression at E13.5, E15.5, E17.5 and P2. Further to the well-established mitogenic roles, IGF2 has anti-apoptotic actions in the endocrine pancreas during the neonatal life [7,24]. Future systematic analyses based on pancreas stereology and measurements of cell proliferation and apoptosis are therefore required to fully understand if IGF2 actions occur in the early mesenchyme and/or mesenchyme-derived cells in later fetal and perinatal development, and the extent to which it influences the differentiation programmes of progenitor cells into exocrine, endocrine and mesenchyme lineages. The genetic models we use in this study have some limitations as well. The effects of both *H19*DMD$^{fl/+}$; *Nkx3.2*$^{+/Cre}$, and *Igf2r*$^{fl/+}$; *Nkx3.2*$^{+/Cre}$ models are not exclusively mediated through IGF2 signalling–*H19*DMD$^{fl/+}$; *Nkx3.2*$^{+/Cre}$ results in downregulation of *H19* ncRNA and its associated exonic *miR-675* [47,48], and *Igf2r* serves as a mannose-6-phosphate receptor in addition to acting as an IGF2 clearance receptor/signalling through G proteins [33,49]. Importantly, the complete rescue of the overgrowth phenotypes due to deletions of maternal *H19* or *Igf2r* alleles in an *Igf2*-null background [50,51] shows that the majority of the effects are mediated through IGF2 signalling. However, we cannot exclude that subtle effects caused by IGF2-independent actions, may explain some of the inconsistent observations regarding mesenchyme and beta-cell mass between these models (Fig 3, Fig 4).

In summary, we report in this study that IGF2 growth actions in the pancreas are limited to the highly expressing mesenchyme-derived cells, and that size-determining mechanisms are programmed in early life by the activity of this important growth factor in a paracrine fashion, with consequences for post-natal physiology. In general, this work provides novel insights into how growth factors control pancreatic architecture and is highly relevant to pancreas pathologies.

## Materials and methods

### Ethics statement

This research has been regulated under the Animals (Scientific Procedures) Act 1986 Amendment Regulations 2012 following ethical review by the University of Cambridge Animal Welfare and Ethical Review Body (AWERB). All mouse experiments were performed under PPL No. 80/2484 (study plan 2484/25/13) and PPL No. 70/7594 (study plan 7594/4/15).

### Generation of the *Igf2*$^{fl/+}$ mouse

The *Igf2* gene targeting vector carried a loxP site inserted 5' of exon 4 and a loxP-flanked neomycin resistance cassette (neoR) inserted 3' of exon 6 (S4A Fig). Details of the cloning procedures are available upon request. In brief, we used a 3.1-kb *Sal*I-*Pac*I genomic fragment (from exon 2 to intron 3 of *Igf2*) as the 5' region of homology (5'-ROH), a 5.8-kb *Pac*I-*Afe*I genomic fragment that includes the entire *Igf2* coding sequence (exons 4 to 6) as internal ROH (Int-ROH), and a 3.1-kb *Afe*I-*Blp*I genomic fragment (3' of exon 6) as 3'-ROH. The Int-ROH was flanked by a 5' loxP site and a 3' loxP-neoR-loxP sequence (S4A Fig). The targeting vector was linearized at a unique *Sca*I site outside the area of homology and 50 μg linearized vector were electroporated into passage 9, E14 129ola male ES cells, at 250V and 950 μF. Transfected cells were plated onto 10 gelatinized 100-mm dishes pre-seeded with fibroblast feeder cells. After 24 h in nonselective medium, cells were incubated for 8 days with G418 medium (200 μg/μl) to select for neomycin resistance. Resistant clones were picked at day 9 and expanded into

96-well plates pre-seeded with fibroblast feeder cells. We initially screened 384 G418-resistant clones by Southern blotting analysis of genomic DNA (gDNA) digested with *EcoR*I and hybridized the blots with a unique 511 bp 5' probe (located external to 5'-ROH and obtained by PCR amplification using primers 5'Pr-F: 5'-AACAACGCGGTGGTAGGGAA-3' and 5'Pr-R: 5'-TCAGCAGAAAAAGAAGCAGGGC-3'). Two correctly targeted clones at the 5' end were then verified by Southern blot (*EcoR*I digested DNA) using a 590 bp 3' probe (located external to 3'-ROH and obtained by PCR amplification using primers 3'Pr-F: 5'-ACAAAGCC CAAGACAACTCC-3' and 3'Pr-R: 5'-CTTCCACAGTTCAAGCAACC-3') and an additional check for multiple integrations elsewhere in the genome using a 583 bp Internal probe (located in *Igf2* exon 6 and obtained by PCR amplification with primers Int-F: 5'-AGAACCCAAGAA GAAAGGAAG-3' and Int-R: 5'-AGAAAGACAGAACTAGCAGCC-3'). One clone with a single integration site and correctly targeted 5' and 3' loxP sites was thus identified (S4A Fig), with the loxP sequences further verified by Sanger sequencing. The neoR cassette was excised by transiently transfecting this ES cell clone with Cre recombinase (pMC-Cre vector), followed by two rounds of subcloning. Correctly excised clones that carry a single 3' loxP site were verified by PCR screening (480 ES subclones) and gDNA digestion with *Sph*I restriction enzyme, followed by Southern blot analysis using the same 583 bp Internal probe described above (S4C–S4F Fig). Primers used for PCR screening of *in vitro* neoR deletion were: F0 5'-TGAC CTCAGCAATTCAAGTCC-3'; F1 5'-GGTAGTGGTCTTTGGCATCC-3' and R1 5'-CAAT AACTGGGGAAAAGGAGC-3'. Two independent ES clones were then microinjected into C57BL/6J blastocysts and transferred into (C57BL/6J X CBA/Ca)F1 pseudo-pregnant females to generate chimeric mice. 27 chimeras were born, all males. Germline transmitting mice were backcrossed into the C57BL/6J genetic background for more than 10 generations before being used as experimental animals. Paternal transmission of the *Igf2* floxed allele did not have any impact on fetal and placental growth kinetics (S4G and S4H Fig). Homozygous *Igf2*^fl/fl mice were obtained by breeding heterozygous *Igf2*^fl/+ parents.

## Mouse strains

Targeted ES cells for the *miR-483* knock-out [31] were generated and provided by Dr. Haydn Prosser (Sanger Institute and International Knockout Mouse Consortium Project). Details on the generation of the *miR-483* knock-out mice (depicted in S7 Fig; Sekita *et al.*, manuscript in preparation) are available upon request. *Rosa26YFP-stop*^fl/fl mice [26] were kindly provided by Dr. Martin Turner (The Babraham Institute, Cambridge). *H19DMD*^fl/fl mice [32] and *Igf2r*^fl/fl mice [52] were generously provided by Prof. Bass Hassan (University of Oxford). *CMV*-Cre mice [53] were obtained from the Babraham Institute, Cambridge. This Cre recombinase is expressed soon after fertilization and allows ubiquitous deletion of floxed alleles in all tissues, including the germline [53]. *Nkx3.2*-knock-in-Cre mice [27] were kindly provided by Dr. Warren Zimmer (Texas A&M University). *Nkx3.2* expression occurs in the mesenchyme of the developing pancreas, stomach and gut, as well as in the forming somites, but not in the endoderm-derived cells of these organs. *Nkx3.2* is expressed in the pancreatic mesenchyme as early as E9.5 and, by E12.5, its expression becomes restricted to the mesenchymal area, which will give rise to the splenic bud [27]. The *Nkx3.2*-knock-in-Cre, in which a Cre recombinase cDNA cassette and a PGK-NeoR cassette were inserted in frame within exon 1 of *Nkx3.2* (*Bapx1*) gene, faithfully replicates endogenous *Nkx3.2* expression and directs Cre activity to the foregut mesenchyme and skeletal somites starting at E9.5 [2,27]. We verified the pancreatic cell-type specificity of *Nkx3.2*-Cre expression using the *Rosa26YFP-stop*^fl/fl reporter mice and demonstrated absence of recombinase activity in acinar cells, pancreatic endocrine cells, endothelial cells or duct cells (Fig 1A) at postnatal day P2. *Tek-Cre* transgenic mice, in which Cre

expression is controlled by the endothelial-specific receptor tyrosine kinase *Tek* (*Tie2*) promoter/ enhancer and therefore found uniformly in endothelial cells during embryogenesis (E7.5 onwards) and adulthood [30], were imported from the Jackson Laboratory (Maine, USA). *Ptf1a*-knock-in-Cre mice, in which the protein-coding region of *Ptf1a* (exon 1 and 2) was replaced with a Cre-cassette, express Cre in pancreatic ducts, exocrine and endocrine cells as early as E10.5 [28]. In newborn pups *Ptf1a*-Cre expression marks all acinar cells and roughly 95% of ductal and insulin-producing cells, as well as 75% of glucagon-producing cells [28]. *RIP-Cre* mice, carry a Cre transgene under the control of the rat *Ins2* (insulin 2) promoter (RIP) that directs expression to insulin-positive beta-cells from approximately E8.5–9 onwards [29]. *Ptf1a*-Cre and *RIP-Cre* strains were obtained from Central Biomedical Services (CBS Transgenic Services, University of Cambridge). All lines were bred into an inbred C57BL/6J line for >10 generations.

## Mouse crosses and genotyping

Mice were fed standard chow diet with 9% of kcal from fat (SDS, Essex, UK) and housed with a 12-h light/dark cycle in a temperature-controlled room (22˚C). For timed mating, the day of detection of a vaginal plug was noted as embryonic day 1 (E1) and the day of birth was noted as post-natal day 0 (P0). Mice were weaned at 3 weeks of age and ear notches were used for visual identification and to collect tissue for PCR genotyping.

Mouse crosses used for each experiment are listed in S1 Table. Throughout the paper, $Igf2^{+/fl}$ represents genotype of offspring that inherited the floxed allele from father; $Igf2^{fl/+}$ represents genotype of offspring that inherited the floxed allele from mother; $Igf2^{+/+}$ represents genotype of offspring with wild-type alleles at the *Igf2* locus; *Nkx3.2*-Cre represents genotype of offspring that are heterozygous Cre-carriers ($^{Cre/+}$ or $^{+/Cre}$); $Igf2^{+/fl}$; $Nkx3.2^{Cre/+}$ represents offspring with mesenchymal *Igf2* deletion on the paternal allele, having inherited the floxed allele from father and the Cre allele from mother; $Igf2^{fl/+}$; $Nkx3.2^{+/Cre}$ represents offspring with mesenchymal *Igf2* deletion on the maternal allele, having inherited the floxed allele from mother and the Cre allele from father; $H19DMD^{fl/+}$; $Nkx3.2^{+/Cre}$ represents offspring with mesenchymal *H19DMD* deletion on the maternal allele, having inherited the floxed allele from mother and the Cre allele from father. All strains were genotyped by standard PCR using DNA extracted from ear biopsies (adult mice) or tail DNA (embryos or post-terminal). PCR was performed using the Red Taq Ready PCR system (Sigma Aldrich) using primers described in S2 Table, followed by separation of PCR amplicons by agarose gel electrophoresis.

## Southern blotting screening of ES clones

gDNA were digested with *EcoR*I or *Sph*I restriction enzymes and electrophoresed on 0.8% agarose gels in 1×TBE buffer, alkaline blotted onto Hybond N+ membranes (Amersham), and UV cross-linked (Stratalinker, Stratagene). Probes were obtained by PCR and radiolabeled (α-32P-CTP). Hybridisation and washing of Southern blots was performed as described [54]. Membranes were exposed overnight to MS film (Kodak).

## Northern blotting analysis of *Igf2* mRNA

Total RNA (10 μg) extracted from E19 placenta and liver samples using RNeasy midi kits (Qiagen) according to manufacturer's protocol, was separated in low-percentage formaldehyde gels, blotted onto Nytran-plus membrane (Schleicher and Schuell), and UV cross-linked (Stratalinker, Stratagene). The RNA blots were hybridized with radiolabelled (α-32P-UTP) *Igf2* and *Gapdh* cDNA probes. We carried out hybridization and washing of northern blots as described [55]. Transcript levels were quantified by PhosphorImager analysis (Molecular analyst software, Biorad).

## Western blotting analyses

For Western blot analysis of IGF2, 5 μl serum samples collected at E19 were loaded into Bis-Tris gels (NuPAGE Novex, Life technologies) and transferred to nitrocellulose membranes (Invitrogen). 4 ng human recombinant IGF2 (292-G2-050, R&D Systems) was used as positive control, and the Novex Sharp protein ladder (Life technologies) was used as marker. The membranes were initially stained with Ponceau red (Invitrogen) and imaged with the Biorad GelDoc system, the intensity of bands being used as internal control for protein loading. The membranes were then blocked with 5% skimmed milk in TBS-T for 1 h at 4˚C, after which were incubated overnight at 4˚C with goat anti-human IGF2 antibody (R&D systems AF-292) diluted 1: 1,000 in TBS-T containing 0.3% skimmed milk. After 4x10 min washes with milliQ water, the blots were incubated for 1h at room temperature with the secondary antibody (1:2,000 rabbit anti-goat IgG coupled to HRP, SantaCruz) in TBS-T containing 3% skimmed milk. Blots were washed 4x10 min with milliQ water, exposed to substrate (Clarity ECL Western Blotting Substrate, Biorad) for 5 minutes and imaged with the Biorad GelDoc system. Bands were quantified using the ImageLab software (Biorad), the output values being normalized to the corresponding Ponceau red loading control values.

For Western blot analysis of AKT phosphorylation, acinar cell pellets were lysed in 30μl RIPA containing protease and phosphatase inhibitors. Protein concentration was determined by DC assay (Biorad) and 10μg/lane total cell lysates were electrophoresed through 4–12% Bis-Tris NuPAGE gels with MOPS running buffer. Proteins were transferred to nitrocellulose by iBlotII (Invitrogen) prior to being blocked for an hour in 3%BSA/TBST. Primary antibody to pAKT (S473) (CST 9271), total AKT (CST 2920), or Beta-actin (CST 4967) was incubated overnight at 4˚C. Membrane was washed 5 times before detection of bound antibody by chemiluminescent signal associated with HRP-conjugated anti-rabbit or anti-mouse secondary antibody.

## Pancreas immunostainings, imaging and stereology analysis

Pancreases were dissected under microscope (for embryonic and early post-natal analyses), fixed in 4% paraformaldehyde in PBS overnight, dehydrated and then embedded in paraffin. Paraffin blocks were cut at 5 μm thickness, sections were then deparaffinised, rehydrated, stained and mounted with coverslips. For all stereological analyses, in order to obtain accurate morphometric estimations without measuring more than once the same feature (e.g. pancreatic islands), sections spaced at 100 μm (P2) or 200 μm (pregnant females) were stained (6–8 sections/block for P2 and 20–24 sections/block for pregnant females). Conditions used for antigen retrieval, blocking and combinations of primary and secondary antibodies used for each staining are described in S3 Table. Counterstaining was performed with haematoxylin for light microscopy stains or DAPI for fluorescent stains.

Light microscopy stained slides were imaged using the NanoZoomer whole slide scanner (Hamamatsu). Fluorescent immunostainings were imaged with a LSM510 Meta confocal laser scanning microscope and the ZEN 2009 software (Carl Zeiss, Germany) or scanned using an Axio whole slide scanner. Whole slide scans of stained sections were analysed using the Visiopharm automated image quantification software (Visiopharm, Denmark) or HALO AI (Indica Labs) to measure the area of positive cell-specific staining or to count individual nuclei. The mass of each cell-type was calculated using the formula:

$$\frac{(area\ of\ cell-specific\ stain\ (\mu m^2)) \times (pancreas\ weight\ (mg))}{(overall\ tissue\ surface\ (\mu m^2))} = mass\ of\ cell\ type\ of\ interest\ (mg)$$

## mRNA in situ hybridization (ISH) for *Igf2*

ISH was performed as described [56], with minor modifications. Briefly, a region of 415 bp spanning coding *Igf2* exons 4–6 was amplified by PCR using primers: F: 5'-CACGCTTCAG TTTGTCTGTTCG-3' and R: 5'-GCTGGACATCTCCGAAGAGG-3' and cDNA from whole P2 pancreases as template. The PCR product was cloned into a pCR2.1-TOPO plasmid (Invitrogen) containing M13 primers and a T7 RNA polymerase transcription initiation site. Sense (S) and antisense (AS) RNA probes were generated and labelled with Digoxigenin (DIG) by *in vitro* reverse transcription, according to manufacturer's instructions (Roche). Pancreases were dissected in cold phosphate buffered saline (PBS) and fixed overnight in 4% paraformaldehyde in 0.1% diethylpyrocarbonate (DEPC)-PBS at 4˚C. After rinsing in DEPC-PBS, tissues were dehydrated and then embedded in paraffin in RNase-free conditions. Pancreas sections (7 μm thick) mounted on polysine slides (VWR) were de-waxed, rehydrated in PBS, post-fixed in 4% paraformaldehyde for 10 minutes, treated with proteinase K (30 μg/ml) for 10 min at room temperature, acetylated for 10 minutes (acetic anhydride, 0.25%) and hybridized overnight at 65˚C in a humidified chamber with DIG-labeled probes diluted in hybridization buffer (containing 200 mM sodium choride, 13 mM tris, 5 mM sodium phosphate monobasic, 5 mM sodium phosphate dibasic, 5 mM EDTA, 50% formamide, 10% dextran sulfate, 1 mg/ml yeast tRNA and 1× Denhardt's [1% w/v bovine serum albumin, 1% w/v Ficoll, 1% w/v polyvinylpyrrolidone]). Two 65˚C post-hybridization washes (1× SSC, 50% formamide, 0.1% tween-20) followed by two room temperature washes in 1× MABT (150 mM sodium chloride, 100 mM maleic acid, 0.1% tween-20, pH7.5) were followed by 30 minutes RNAse treatment (400 mM sodium chloride, 10 mM tris pH7.5, 5 mM EDTA, 20 μg/ml RNAse A). Sections were blocked for 1 hour in 1×MABT, 2% blocking reagent (Roche), 20% heat inactivated goat serum and then incubated overnight with anti-DIG antibody (Roche; 1:2,500 dilution) at 4˚C. After 4x20 min washes in 1× MABT, slides were rinsed in 1× NTMT (100 mM NaCl, 50 mM MgCl, 100 mM tris pH 9.5, 0.1% tween-20) and incubated with NBT/BCIP mix in NTMT buffer, according to manufacturer's instructions (Promega). Slides were counterstained with nuclear fast red (Sigma), dehydrated and cleared in xylene and mounted in DPX mounting medium (Sigma). Pictures were taken with a camera attached to a light microscope.

## Fluorescence-activated cell sorting (FACS)

Mice were sacrificed by decapitation (embryos and P2 neonates) or cervical dislocation (P5 neonates and older). Then, pancreases were dissected and dissociated into single cells using trypsin-EDTA (Sigma Aldrich) at 37˚C for 20 min. After one wash with ice-cold PBS, the cells were passed through 70 μm strainers and single-cell suspensions were sorted into YFP-positive and YFP-negative fractions using an Aria-Fusion cell sorter (BD Bioscience). Dead cells were excluded based on forward and side scatter profiles and the uptake of 7AAD (7-Aminoactinomycin D dead cell stain, Life Technologies). Sorted YFP-positive and YFP-negative cells were pelleted by centrifugation and flash frozen in liquid nitrogen, then stored at -80˚C until use.

## Total RNA and microRNA extraction (pancreas and sorted cells)

'Total RNA was extracted from whole pancreases, cells isolated by FACS or cultured cells using the RNEasy Mini kit (Qiagen) or the RNEasy Micro kit (Qiagen), respectively, according to manufacturer's instructions, with additional removal of contaminating DNA using QIASpin DNA eliminator columns. microRNA was extracted using the miRNeasy kit (Qiagen), according to manufacturer's instructions with an additional step of DNaseI treatment (Qiagen).

## Quantitative real-time PCR (qRT-PCR)

Concentration and integrity of total RNA was verified by NanoDrop (Thermo Scientific) and agarose gel electrophoresis, respectively. For RNA extracted from whole pancreases, reverse transcription was performed using the RevertAid RT Reverse Transcription Kit (Life technologies), according to manufacturer's instructions. In the case of total RNA extracted from sorted cells, cDNA was produced using the QuantiTect Whole Transcriptome Kit (Qiagen) following manufacturer's instructions. Primers used for qRT-PCR are listed in S4 Table. Annealing temperatures were tested by gradient PCR using pancreas-cDNA as a template. qRT-PCR was performed using the SYBR Green JumpStart Taq Ready Mix (Sigma Aldrich) on an ABI Prism 7900 system (Applied Biosystems). Gene expression levels were normalized to the housekeeping gene *Ppia* (peptidylpropyl isomerase A or cyclophilin-A), which has previously been established as a good housekeeping gene for pancreas gene expression studies [57] and is stably expressed between various developmental time points. For microRNAs, reverse transcription was performed using the TaqMan Micro RNA Reverse Transcription kit (Applied Byosystems), according to manufacturer's protocol. qRT-PCR was performed using TaqMan assays (TM: 002560 for mmu-miR-483; TM: 001232 for snoRNA202 and TM: 001234 for snoRNA234) and TaqMan 2x Universal PCR Master Mix (Applied Byosystems). Levels of expression were calculated using the $2^{-\Delta\Delta Ct}$ method [58].

## Complementary DNA library preparation and RNA-seq analysis

Input RNA for genome-wide transcriptome analysis was verified for concentration and quality using Agilent RNA Pico chips, according to manufacturer's instructions. All RNA samples had RNA integrity numbers (RIN) >7.5. Total RNA (2 ng) was whole-transcriptome amplified using the Ovation RNA–seq System V2 (NuGEN). To prepare the RNA–seq libraries, the amplified cDNA (2 µg per sample) was fragmented to 200 bp using a Bioruptor Sonicator (Diagenode), and barcode ligation and end repair were performed using the Ovation Rapid DR Library System (NuGEN). The barcoded libraries were combined and loaded onto an Illumina HiSeq 2500 system for single-end 50-bp sequencing at the Genomics Core Facility, Cambridge Institute, CRUK. The reads were aligned onto the mouse GRCm38 genome using TopHat 2.0.11 [59]. Gene abundance and differential expression were determined with Cufflinks 2.2.1 [60] and expressed in fragments per kilobase per million mapped reads (FPKM). The cut off for expression was set at ≥ 1 FPKM. Genes with a linear fold expression change greater than 1.5 and a Benjamini–Hochberg false discovery rate <5% were considered differentially expressed.

## Functional annotation and enrichment analysis

DAVID (Database for Annotation, Visualization and Integrated Discovery; v6.8 http://david.abcc.ncifcrf.gov/, accessed March 2017) was performed to assess whether there was enrichment for genes implicated in particular biological processes within the differential expressed gene lists identified by RNA-seq. Enriched gene ontology (GO) terms with an FDR < 0.05 were considered significant. These terms were then clustered semantically using REViGO (Reduce and Visualize GO) [61], which removes redundancy. The results obtained by REViGO were ordered according to log10 p values.

## Whole pancreas lipase content and amylase measurements

For whole pancreas lipase measurements, frozen samples were removed from -80°C and thawed on ice. 100 µL TK lysis buffer with proteinase inhibitors (Calbiochem) was added to

each pancreas and tissues were disrupted on ice with a homogenizer (Biospec Tissue Tearor). Lysates were spun at 3,000 rpm for 15 min at 4°C and supernatants were used for analysis. Lipase levels were measured using a lipase activity assay (Dimension RXL, Siemens).

For amylase measurements, cultured acinar cells were lysed in 100 μL RIPA buffer, spun at 3,000 rpm for 15 min at 4°C and the supernatants used for analysis. Amylase levels were measured in an autoanalyser (Siemens Dimension RXL) through a colorimetric reaction based on the ability of amylase to hydrolyse the chromogenic substrate 2-chloro-4-nitrophenol linked with maltotriose (CNPG3) into the coloured product 2-chloro-4-nitrophenol (CNP) plus CNPG2, maltotriose G3 and glucose. The absorbance increase at 405 nm is proportional to the amylase activity in the sample. Amylase levels (U/L) were normalized to the total protein content determined by a BCA assay (Pierce BCA Protein Assay Kit, Thermo Fisher Scientific, 23225).

## Primary pancreatic acinar and mesenchymal cell isolation and culture

Primary pancreatic cells were isolated as previously described [62] and adapted here to P2 samples. Briefly, pancreases from an entire P2 litter were micro-dissected under microscope and sterile conditions and pooled in one tube containing HBSS, 0.25 mg/ml of trypsin inhibitor and 1% Penicillin-Streptomycin mix. After rinsing, pancreases were digested for 20–30 min at 37°C with a collagenase IA solution (HBSS containing 10mM HEPES, 200 U/ml of collagenase IA, and 0.25 mg/ml trypsin inhibitor). The digestion was stopped by placing the samples on ice and addition of FBS (fetal bovine serum) for a 2.5% final concentration. After additional washing steps, the cell suspension was passed through a 100 μm cell strainer, allowing the passage of acini and mesenchymal cells, while retaining the non-digested fragments and larger pancreatic islets. The cells were placed in the culture media (basal Waymouth's media containing 2.5% FBS, 1% Penicillin-Streptomycin mix, 0.25 mg/ml trypsin inhibitor, and 25 ng/ml of recombinant human EGF) and plated into six-well plates pre-coated with poly-L-lysine. After one hour incubation at 37°C under 5% (v/v) $CO_2$ atmosphere, the floating acini were transferred to a new well, while the mesenchyme cells remained attached. The cells were cultured under the above conditions for up to four days, with daily media changes.

## Measurement of IGF2 secretion by primary mesenchymal and acinar cells *in vitro*

The primary mesenchymal and acinar cells were isolated and cultured as described above until reaching confluency. Media (1mL) was collected 24h post-confluency and freeze-dried using a Christ Gamma 2–16 LSC Freeze dryer. Dry pellets were re-dissolved in 50 μl RIPA buffer and IGF2 measured by ELISA (Mouse IGF-II DuoSet ELISA kit, R&D Systems–DY792) using an assay adapted for the MesoScale Discovery electrochemiluminescence immunoassay platform (MSD), as recently described [63].

## *In vitro* IGF2 treatments

To assess the impact of exogenous IGF2 on intracellular signalling via AKT, freshly isolated acinar cells were starved for two hours in basal Waymouth's media, without FBS and EGF. IGF2 treatments (with vehicle or 1 nM, 10 nM or 50 nM mouse recombinant IGF2) were performed for 10 min at 37°C under 5% (v/v) $CO_2$ atmosphere, then the cells were immediately placed on ice, washed two times in ice-cold PBS, pelleted and flash-frozen on dry-ice and stored at -80°C until analysed by Western blotting.

To assess the impact of exogenous IGF2 on amylase production by acinar cells, subconfluent acinar cultures (48 hours after isolation) were placed in basal Waymouth's media containing 1% Penicillin-Streptomycin mix and 0.25 mg/ml of trypsin inhibitor. FBS was replaced

with Serum Replacement (Sigma Aldrich S0638) that does not contain any IGF2. The treated cells received 50 ng/ml mouse recombinant IGF2 for 48 hours, with new media added every day. After the treatment, cells were harvested and dissociated into single-cell suspension using trypsin, counted with a Cedex XS Analyser (Roche), washed, pelleted and stored at -80˚C until used for amylase measurement.

### Oral glucose tolerance test

Conscious mice were used for an oral glucose tolerance test after 6 hours of fasting (from 8am to 2pm). Blood samples ($\leq$5 μL) were taken from the tail vein immediately before administration of glucose by oral gavage (20% weight for volume, 2g/kg body weight based on average body weights for each experimental group) and thereafter at the time points indicated, and used for glucose measurements with a glucose meter (AlphaTRAK).

### Statistical analysis

Numerical data used to generate all graphs presented in this study are shown in S5 Table. Statistical analysis was performed using GraphPad Prism 7 software. For two groups with up to 6 samples, statistical analysis was performed using Mann-Whitney or the Wilcoxon matched-pairs signed rank test. For two groups with $\geq$6 samples, to determine whether the data was of a Gaussian distribution, the Shapiro-Wilk test was first applied, followed by un-paired or paired Student's t-tests, as appropriate. Where more than two groups exposed to the same treatment were analysed, 1-way ANOVA with Dunnett's post-hoc or Friedman's test with Dunn's correction tests were used, comparing every mean to a control mean. OGTT data was analysed using two-way ANOVA with Sidak's multiple comparison tests, using the time from administration and the genotype as factors (two genotypes) or by repeated measures 1-way ANOVA with Dunnett's post hoc test for multiple comparisons against the controls (four genotypes). The area under the curve (AUC) was calculated by the trapezoidal rule and used for analyses by unpaired Student's t tests (two genotypes) or by 1-way ANOVA with Dunnett's post-hoc tests (four genotypes). Unless stated otherwise, data is shown as average values and error bars represent SEM or SD. N.S. $p > 0.05$; * $p \leq 0.05$; ** $p \leq 0.01$, *** $p \leq 0.001$.

## Supporting information

**S1 Data. List of genes significantly enriched (fold change >1.5; FDR adjusted p value <0.05) in mesenchyme and non-mesenchyme pancreatic cells at P2 in wild-type mice (Igf2 +/+; Rosa26YFP-stop+/fl; Nkx3.2Cre/+) and associated functional pathway analyses (DAVID).**
(XLSX)

**S2 Data. List of DEGs (fold change >1.5; FDR adjusted p value <0.05) in non-mesenchymal cells between Igf2+/fl; Rosa26YFP-stop+/fl; Nkx3.2Cre/+ knockouts and Igf2+/+; Rosa26YFP-stop+/fl; Nkx3.2Cre/+ controls at P2, and associated functional pathway analyses (DAVID).**
(XLSX)

**S3 Data. List of DEGs (fold change >1.5; FDR adjusted p value <0.05) in mesenchyme-derived cells between Igf2+/fl; Rosa26YFP-stop+/fl; Nkx3.2Cre/+ knockouts and Igf2+/+; Rosa26YFP-stop+/fl; Nkx3.2Cre/+ controls at P2, and associated functional pathway analyses (DAVID).**
(XLSX)

**S1 Fig. Assessment of efficiency of pancreas-expressing Cre lines using the *Rosa26YFP*-stopfl/fl reporter mouse, at postnatal day 2 (P2).** (A) Strategy used for sorting YFP+ and YFP- cells by FACS. Top left panel: side scatter area (SSC-A) versus forward scatter area (FSC-A) plotting was used to identify cells of interest (gated) based on granularity and size, respectively. Top right panel: forward scatter width (FSC-W) versus forward scatter area (FSC-A) was then used to exclude cell doublets. Bottom panels: gates used to discriminate YFP + (green and blue rectangles) versus YFP- cells (red rectangle), after exclusion of dead cells (positive for 7-Aminoactinomycin D [7-AAD]) in cells isolated from pancreases of two crosses: *Rosa26YFP*-stop$^{+/fl}$; Nkx3.2$^{Cre/+}$ that activates YFP in the mesenchyme-derived cells (bottom left panel) and *Rosa26YFP*-stop$^{+/fl}$; *RIP*$^{Cre/+}$ that activates YFP in the pancreatic beta-cells (bottom right panel). (B) *Igf2* mRNA expression measured by qRT-PCR in YFP+ cells collected from offspring with the genotypes indicated. Expression data was normalized to *Ppia* and shown as averages + SEM relative to levels measured in control (C) littermates, arbitrarily set to 1. Percentage values indicate the level of *Igf2* mRNA reduction relative to controls (n = 6 samples/genotype; $^{**}$ p<0.01 by Mann-Whitney tests).
(TIF)

**S2 Fig. Gene expression patterns of pancreatic mesenchyme-derived cells and non-mesenchyme cells identified by RNA-seq at postnatal day 2 (P2).** (A) Volcano plot representing all expressed transcripts in pancreatic mesenchyme and non-mesenchyme cells (purified by FACS). For every transcript, the Log$_2$ fold change between mesenchyme and non-mesenchyme cells was plotted against the -Log$_{10}$ p value (FDR adjusted). Genes significantly enriched, with a fold change >1.5 and FDR adjusted p value <0.05, in mesenchyme are depicted as blue dots (n = 1,902 genes), in non-mesenchyme as red dots (n = 2,212), and those not significantly enriched in either cell fraction are depicted as black dots (all genes are listed in S1 Data). Selected genes of interest, such as known mesenchyme-expressed genes, signature genes for the various non-mesenchyme cell-types and imprinted genes are grouped by colour. (B) Biological validation by qRT-PCR of DEGs between mesenchyme-derived cell and non-mesenchyme cells, identified by RNA-seq (n = 5–6 samples per group). Expression levels were normalized to *Ppia*. Data is shown are average values; error bars represent SEM; $^{**}$ p<0.01 by Mann-Whitney tests. (C) Top scoring biological processes containing genes enriched in the pancreatic non-mesenchyme cells as identified by DAVID functional annotation. The dotted line in panel corresponds to a p value of 0.05. (D) Top 30 expressed genes with highest FPKM values in pancreatic non-mesenchyme cells (average FPKM + SEM in mesenchyme and non-mesenchyme cells with n = 4 per group) at P2. Note that all 30 genes are significantly enriched in non-mesenchyme compared to mesenchyme-derived cells (>1.5 fold, FDR adjusted p value <0.05).
(TIF)

**S3 Fig. Expression of marker genes in mesenchyme-derived cells by RNA-seq.** (A) Mesenchyme-derived cells exhibit high expression levels for known markers of pericytes, smooth muscle cells, fibroblasts and myofibroblasts. Expression levels (shown as FPKM + SEM) were calculated by analysing the RNA-seq data obtained in E13.5, E15.5 and E17.5 mesenchyme-derived cells (n = 2 for each developmental time point) by Harari N *et al.* [3]. (B) Expression levels in P2 mesenchyme-derived cells for the same markers genes shown in panel (A) (n = 4; data shown as FPKM + SEM).
(TIF)

**S4 Fig. Generation of *Igf2* conditional knockout mice.** (see also Materials and Methods for further details). (A) Targeting strategy to generate an *Igf2* allele with coding exons 4 to 6

flanked by loxP sites (not drawn to scale). Blue boxes–exons; P0-P3 –alternative promoters; green triangles–loxP sites; neo–neomycin cassette; WT–wild-type; T–targeting (construct); E–EcoRI restriction sites; 5', Int, 3': location of 5', internal and 3' Southern blotting probes, respectively (B) Southern blot confirmation of homologous recombination between targeting vector and endogenous *Igf2* sequences in genomic ES cell DNA, digested with EcoRI and hybridized with 5', Int or 3' probes. Diagnostic molecular weights (kb) are indicated in each panel. T and WT–targeted and wild-type clones, respectively. (C) Screening strategy for loxP recombination events (not drawn to scale). Correctly targeted ES clones (as shown in (B)) that are transiently exposed to Cre recombinase *in vitro* will undergo three possible independent recombination events involving the loxP sites, which can be discriminated by PCR and Southern blotting. F0, F1, R1 –PCR screening primers; S–SphI restriction sites; Int–Internal Southern probe (D) Five 96-well plates containing targeted ES cells transfected with Cre recombinase were screened by F1+R1 primer PCR (panel i) or F0+R1 primer PCR (panel ii). F1+R1 PCR products of 221 bp, 1.4 kb and 328 bp are diagnostic of the wild-type allele, neomycin cassette, and neomycin cassette deletion, respectively (panel i). F0+R1 PCR products of 520 bp are diagnostic of a deletion that includes the neomycin cassette and exon4-6 region (in panel (ii)). Since all clones that had deletion of the neomycin cassette (*i.e.* 328 bp) also had cells with deletion of exon4-6 region (520 bp), the clone indicated with a star in panel (i) was subsequently subcloned for selection of cells with neomycin cassette excision events only. (E) Representative ES subclones (starred) with deletion of the neomycin cassette only (*i.e.* 328 bp in panel (i) but absence of 520 bp PCR product in panel ii). (F) Southern blot confirmation of deletion of the neomycin cassette in ES subclones identified in (E). DNA was digested with SphI and hybridized with the internal probe: 6.4 kb–wild-type allele (WT); 3.9 kb–floxed allele (fl), resulting from Cre-induced deletion of the neomycin cassette. (G) PCR genotyping of *Igf2*$^{+/fl}$ mice was performed using primers F and RW flanking the 5' loxP site (primer sequences are shown in S2 Table). A representative example of tail DNA PCR is shown for mice carrying one floxed *Igf2* allele (lane 1) wild-type littermate (lane 2); lane 3 –no template PCR control; lane 4–100 bp DNA ladder. (H) Carriers of an *Igf2* floxed allele that was inherited paternally show identical embryonic and placental weights as littermate controls (E12: n = 8 wild-type and n = 3 floxed; E14: n = 16 wild-type and n = 7 floxed; E16 n = 12 wild-type and n = 10 floxed; E19: n = 29 wild-type and n = 24 floxed). Data is shown as average weight; error bars represent standard deviation (SD).
(TIF)

**S5 Fig. *CMV*-Cre-mediated deletion of the *Igf2* floxed allele in vivo.** (A) PCR strategy to identify deletion events at DNA level. PCR products obtained in a tri-primer (F+RW+RD) PCR are diagnostic of the wild-type allele (WT: 324bp), floxed allele (fl: 449 bp) and deleted allele (Del: 384bp). Representative tail DNA PCR examples are shown in (B) for *Igf2*$^{+/fl}$ mice that carry floxed and wild-type alleles (lane 1), *Igf2*$^{+/fl}$; *CMV*$^{Cre/+}$ mice carrying deleted and wild-type alleles (lane 2), *Igf2*$^{+/+}$ control littermates with wild-type alleles only. (C) to (G) *CMV*-Cre mediated deletion of the *Igf2* floxed allele *in vivo*. Heterozygous floxed *Igf2* males were mated with females homozygous for *CMV*-Cre (active in all cells of embryos and placentae), and offspring analysed for levels of *Igf2* deletion by northern blotting (C), Western blotting (D) and (E), and growth curves from E14 to E19 of gestation ((F) and (G)). (C) Northern blot analysis of *Igf2* mRNA levels, showing wild-type levels of expression in controls (*Igf2*$^{+/+}$; *CMV*$^{Cre/+}$) in both placenta (PL) and liver (LV) at E19, and absence of all *Igf2* transcripts upon *CMV*-Cre mediated deletion of the paternally inherited floxed allele (*Igf2*$^{+/fl}$; *CMV*$^{Cre/+}$). *Gapdh*–internal control for RNA loading. (D) Western blotting analysis of the mature form of IGF2 in serum samples collected from E19 controls (C: *Igf2*$^{+/+}$; *CMV*$^{Cre/+}$) and mutant (KO:

*Igf2*$^{+/fl}$; *CMV*$^{Cre/+}$) embryos. Normalisation of IGF2 expression across samples was performed against Ponceau-stained protein band (arrow), and the relative quantification of IGF2 levels for the two genotypes is shown in (E). Data is shown as average values; error bars represent SEM; ** $p < 0.01$ by Mann-Whitney test. (F) and (G) Mice with a *CMV*-Cre mediated deletion of the paternally inherited *Igf2* floxed allele (*Igf2*$^{+/fl}$; *CMV*$^{Cre/+}$) show a similar embryonic (F) and placenta (G) growth phenotype to *Igf2* null mice, i.e. ~ half of the weight of littermate controls (*Igf2*$^{+/+}$; *CMV*$^{Cre/+}$) at the end of gestation (E14: n = 3 litters; E16: n = 9 litters; E19: n = 4 litters). Data is shown as average values; error bars represent SD; * $p < 0.05$; ** $p < 0.01$; *** $p < 0.001$ using paired student t tests.
(TIF)

**S6 Fig. Inheritance of *Nkx3.2*-Cre or *Igf2* floxed alleles, or deletion of maternal *Igf2* allele do not affect total body or pancreas weights at postnatal day 2 (P2).** (A) Body and pancreas weights in offspring obtained from a cross between heterozygous *Nkx3.2*-Cre females and heterozygous *Igf2* floxed males. (B) Body and pancreas weights in offspring obtained from a cross between heterozygous *Igf2* floxed females and heterozygous *Nkx3.2*-Cre males. (C) *Igf2* mRNA levels measured by qRT-PCR in pancreases with a deletion of the maternal *Igf2* allele in the mesenchyme. Data is normalized to *Ppia* and shown relative to average *Igf2* levels in controls (C–*Igf2*$^{+/+}$; *Rosa26YFP*-stop$^{fl/+}$), set to 1. Data is shown as averages or individual values; error bars represent SEM. Numbers shown indicate numbers of animals for each genotype. Data was analysed using one-way ANOVA with Dunnett's multiple comparison test against the control group for panels (A) and (B) and by unpaired Student's t test in panel (C); N.S.–non-significant; *** $p < 0.001$.
(TIF)

**S7 Fig. Paternally inherited *miR-483* deletion does not lead to reduction in pancreas size.** (A) Gene targeting strategy to delete the intronic *Igf2 miR-483*. In brief, a targeting vector was used to replace the *miR-483* sequence in intron 4 of the *Igf2* gene with a PGKPuroΔtk selection cassette in ES cells. Removal of the selection marker was achieved by Cre-recombination between loxP sites. Germline transmitting *miR-483* KO chimeric mice were generated by ES cell injection into blastocysts (Sekita, Prosser, Zvetkova *et al*., manuscript in preparation). (WT–wild-type allele, T–targeted allele, KO–knock-out allele, P0 –P3 are alternative *Igf2* promoters, Ex4 –Ex6 are the *Igf2* coding exons, F and R indicate the position of the forward and reverse primers used for PCR genotyping). (B) *miR-483* and *Igf2* expression levels measured by qRT-PCR in postnatal day 2 (P2) whole pancreas from offspring of heterozygous *miR-483* KO males mated with wild-type females and shown relative (%) to littermate controls (C–*miR-483*$^{+/+}$) set to 1. Expression data was normalized to *snoR-202* and *snoR-234* (for *miR-483*) and to *Ppia* (for *Igf2*) and is shown as average + SEM (n = 10 C and n = 10 KO). N.S.–non-significant; *** $p < 0.001$ by unpaired Student's *t* test. (C) Total body weights and pancreas weights. Data is shown as averages or individual values (n = 33 C and n = 24 KO); error bars represent SEM; N.S.–non-significant differences by two-way ANOVA using genotype and litter as factors.
(TIF)

**S8 Fig. Deletion of paternal *Igf2* in pancreatic mesenchyme leads to reduced body and pancreas weights at weaning (P21).** (A) Total pancreas weights and (B) body weights in offspring obtained from a cross between heterozygous *Nkx3.2*-Cre females and *Igf2* floxed males. The value within brackets in (A) shows % pancreas weight reduction after normalization to body weight. Data is shown as individual values or averages; error bars represent SEM. Numbers of mice for each genotype are shown. Data was analysed using one-way ANOVA with Dunnett's

multiple comparison tests against the control group (C–*Igf2*<sup>+/+</sup>; *Rosa26YFP*-stop<sup>+/fl</sup>); N.S.–
non-significant; * p<0.05; *** p<0.001.
(TIF)

**S1 Table. Mouse strains and crosses.**
(DOCX)

**S2 Table. Primers used for genotyping by PCR.**
(DOCX)

**S3 Table. Conditions used for pancreas immunostaining.**
(DOCX)

**S4 Table. Primers/Assays used for qRT-PCR.**
(DOCX)

**S5 Table. Numerical data underlying graphs shown in this study.**
(XLSX)

## Acknowledgments

We thank the NIHR Cambridge BRC Cell Phenotyping Hub (in particular we wish to thank
Natalia Savinykh for help with flow cytometry cell sorting), Debbie Drage, Martin George and
in particular Ted Saunders (The Babraham Institute Gene Targeting Facility) for help with
generating the *Igf2*<sup>+/fl</sup> mice; Adrian Wayman (West Forvie Phenomics Center) and Adriana
Izquierdo-Lahuerta and Yurena Vivas (Universidad Rey Juan Carlos) for help with mouse hus-
bandry; Keli Philips, James Warner and Katherine Vickers (Histopathology Core) for help
with preparing tissue samples for histology and insulin staining; Gregory Strachan (Imaging
Core facility) for help with confocal microscopy imaging; Keith Burling (Biochemical Assay
Laboratory) for performing lipase, amylase and IGF2 measurements; Dan Hart (MRL Disease
Model Core) for performing oral glucose tolerance tests; Dr. Claire Stocker from the Univer-
sity of Buckingham for providing training on pancreas stereology; Dr. Allan Bradley for pro-
viding *miR-483* knockout ES cells. Schematic representations shown in Fig 3A, Fig 4A, Fig 4E,
Fig 5D, Fig 6A and Fig 6H were generated using BioRender (Biorender.com).

## Author Contributions

**Conceptualization:** Constanze M. Hammerle, Ionel Sandovici, Antonio Vidal-Puig, Susan E.
Ozanne, Gema Medina-Gómez, Miguel Constância.

**Data curation:** Constanze M. Hammerle, Ionel Sandovici, Gemma V. Brierley, Nicola M.
Smith, Brian Y. H. Lam, Wendy N. Cooper, Gema Medina-Gómez, Miguel Constância.

**Formal analysis:** Constanze M. Hammerle, Ionel Sandovici, Gemma V. Brierley, Nicola M.
Smith, Brian Y. H. Lam, Wendy N. Cooper.

**Funding acquisition:** Antonio Vidal-Puig, Susan E. Ozanne, Miguel Constância.

**Investigation:** Constanze M. Hammerle, Ionel Sandovici, Gemma V. Brierley, Nicola M.
Smith, Yoichi Sekita, Brian Y. H. Lam, Wendy N. Cooper, Gema Medina-Gómez.

**Methodology:** Constanze M. Hammerle, Ionel Sandovici, Gemma V. Brierley, Nicola M.
Smith, Ilona Zvetkova, Haydn M. Prosser, Yoichi Sekita, Marcella Ma.

**Project administration:** Miguel Constância.

**Resources:** Warren E. Zimmer, Ilona Zvetkova, Haydn M. Prosser, Yoichi Sekita.

**Software:** Brian Y. H. Lam, Wendy N. Cooper.

**Supervision:** Miguel Constância.

**Validation:** Constanze M. Hammerle, Ionel Sandovici, Gemma V. Brierley, Miguel Constância.

**Visualization:** Constanze M. Hammerle, Ionel Sandovici.

**Writing – original draft:** Constanze M. Hammerle, Ionel Sandovici, Miguel Constância.

**Writing – review & editing:** Constanze M. Hammerle, Ionel Sandovici, Gemma V. Brierley, Brian Y. H. Lam, Susan E. Ozanne, Gema Medina-Gómez, Miguel Constância.

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
