## [Decision Letter · Decision Letter 0]

10 Jun 2020

Dear Dr Sandovici,

Thank you very much for submitting your Research Article entitled 'Mesenchyme-derived IGF2 is a major paracrine regulator of pancreatic growth and function' to PLOS Genetics. Your manuscript was fully evaluated at the editorial level and by independent peer reviewers. The reviewers appreciated the attention to an important topic but identified some aspects of the manuscript that should be improved.

We therefore ask you to modify the manuscript according to the review recommendations before we can consider your manuscript for acceptance. Your revisions should address the specific points made by each reviewer.

[LINK]

Yours sincerely,

David J Hill, D.Phil

Guest Editor

PLOS Genetics

Gregory Barsh

Editor-in-Chief

PLOS Genetics

The reviewers have provided detailed reviews and find the paper potentially suitable for publication in PLOS Genetics. Referee 1 raises a number of issues around the amount of data shown that seems somewhat peripheral to the main message and suggests a shortening of the manuscript. While both reviewers consider that no important data is missing, there are a number of recommendations as to the how the flow and construction of the paper could be improved.

Reviewer's Responses to Questions

**Comments to the Authors:**

Reviewer #1: This study reports that parentally-imprinted genes are highly expressed in pancreatic mesenchyme-derived cells and studies the role of Igf2 in mesenchymal and epithelial pancreatic lineages. A conditional Igf2 mouse model was used to show that mesenchyme-specific Igf2 deletion results in acinar and beta-cell hypoplasia, whole-body growth restriction postnatally and maternal glucose intolerance during pregnancy. This suggests that the mesenchyme is a reservoir of IGF2 throughout life contributing to paracrine control. When Igf2 was deleted in developing pancreatic epithelium. None of the above effects were seen. Increased IGF2 levels specifically in the mesenchyme resulting from conditional Igf2 loss-of-imprinting or Igf2r deletion caused pancreatic acinar overgrowth. Ex-vivo exposure of primary acinar cells to exogenous IGF2 activated AKT and increased cell number and amylase production. Results suggest that mesenchymal Igf2 is a developmental regulator of adult pancreas size and function.

The data is original and the use of transgenic strains innovative to tease out the contributions of mesenchymal Igf2. However, the overall effects of the targeted deletion are quite modest both in beta cell mass with no change in glucose intolerance outside of pregnancy. This probably stems from the fact that IGF2 derives from other cell compartments in the islets also and well as the blood. The paper is something of a data dump with huge swathes of material that are quite peripheral to the central hypothesis. While these data represent a huge amount of work they tend to complicate and detract from the main message of the paper. I think it could be shortened considerably without losing the key experiments.

Below are a number of points that could be addressed:

The latter part of the Introduction would be better placed in the Discussion as this sums up the results.

It is stated that Nkx3.2 expression occurs in the mesenchyme of the developing pancreas, stomach and gut, as well as in the forming somites, but not in the endoderm-derived cells of these organs. Is epithelial to mesenchymal transformation a feature of migration of the endocrine progenitors out of the embryonic pancreatic ductal tree to form the dispersed islets at all? This might complicate the transcription factor identification of mesenchyme vs. endoderm cell types.

Figure 2B. I am not sure how these values might be interpreted. Igf1 is also highly expressed in mesenchyme and given its higher affinity for the Igf1R might be more biologically important. Also, there are numerous Igfbps expressed such that the levels of free Igf2 available for receptor signaling may be much lower than the total Igf2. There is no comparison given for the ectoderm compartment so it is difficult to make comparisons. For Figure 2A how would be pathway analysis differ for any other developing tissue compartment in a rapidly growing animal? How is the pancreatic mesenchymal tissue any different from liver or kidney or lung for instance? The same could be asked for Figure 2C.

Interpretation of a reduced pancreatic weight and beta cell mass in the mesenchymal Igf2 ko mice is solely in terms of the mitogenic effects of IGF2. However, in the developing neonatal pancreas IGF2 is anti-apoptotic and can equally influence cell number by this mechanism.

Within a paracrine environment of the P2 pancreas it is clear that the epithelial cells and endothelial cells are the major relative contributors to IGF2 content as opposed to mesenchyme (Fig 2F). The beta cells will not differentiate between IGF2 derived from any source, including that in the circulation in the young mouse. Again, not sure how this data reinforces the hypothesis.

In Fig 3 the reduction or over-expression strategies for mesenchymal Igf2 expression have clearly worked, but how about the actual numbers of mesenchymal cells in each? If the mesenchymal IGF2 is a growth factor for the mesenchymal cells the reduction or increase in pancreas weight may reflect changes on the mesenchymal cell number. The reduction in beta cell mass in Fig 3D is modest, and no such data is shown for the mesenchymal Igf2 over-expression model.

Figure 4 is not surprising in that if you deleted the expression of any growth factor from a tissue compartment you would see changes in the expression of other classes of genes. What is this specifically indicating about mesenchymal Igf2? Deletion of the mesenchymal Igf2 appears to be invoking the expression of immune system and chemokine genes. Therefore, is the overall level of inflammation altered in these animals, which could explain later phenotypic changes? If the mesenchymal Igf2 ko mice were stressed postnatally through a high fat diet would they become diabetic due to underlying beta cell metabolic stress? This might also underlie the impaired glucose tolerance seen in pregnancy. The pregnancy experiments could be extended as an adaptive 2-3 fold increase in beta cell mass normally occurs in mice by gestational day 19. Was this impaired in the Igf2 ko mice?

Reviewer #2: Review of Mesenchyme-derived IGF2 is a major paracrine regulator of pancreatic 1 growth and function

This manuscript from Hammerle et al reports on the role of mesenchymal IGF2 and IGF2 signaling in the control of pancreas size. This is very compelling data set using multiple lineage specific loss-of-function models, as well as gain-of-function model to make the case the imprinted Igf2 in the mesenchyme plays a critical role.

This is an understudied area and the present results break important new ground.

The data and figures are very well presented. The approach is systematic.

This is an exceptional paper.

Minor comments/questions:

Does the IGF2 act solely via the IGF2R on the acinar cells, or is there a role for INSR or INSR/IGF2R hybrid receptors?

Given the emerging role for insulin and insulin like growth factor signalling on acinar and duct cells in pancreatic cancer (Zhang 2019 Cell Metabolism), could the axis described here be reactivated in adults and contribute to the initiation of pancreatic cancer?

What is the predicted IGF2 levels within the pancreas in adults and how do these compare with the in vitro studies?

Line 383-5: “Although we provide evidence that mesenchymal IGF2 is the long-sought promotor of acinar growth, we have not established the precise timing of the actions of IGF2.” Should be changed to “a long-sought promoter” since it is likely that there is more than one.

In Fig. 5. It would be ideal if the authors could show insulin levels. If there are pancreas sections from this time point, and analysis of beta-cell mass would greatly improve the paper and provide some mechanistic insight into the impaired glucose homeostasis. The authors should elaborate a little more about what they think is underlying this pregnancy-induced defect.

**Have all data underlying the figures and results presented in the manuscript been provided?**

Reviewer #1: Yes

Reviewer #2: Yes

PLOS authors have the option to publish the peer review history of their article (what does this mean?). If published, this will include your full peer review and any attached files.

Reviewer #1: No

Reviewer #2: No

---

## [Editor Report · Decision Letter 1]

20 Aug 2020

Dear Dr Sandovici,

We are pleased to inform you that your manuscript entitled "Mesenchyme-derived IGF2 is a major paracrine regulator of pancreatic growth and function" has been editorially accepted for publication in PLOS Genetics. Congratulations!

Yours sincerely,

David J Hill, D.Phil

Guest Editor

PLOS Genetics

Gregory Barsh

Editor-in-Chief

PLOS Genetics

Comments from the reviewers (if applicable):

The paper has been revised to take account of all of the reviewer comments.This has improved the manuscript.

**Data Deposition**

http://datadryad.org/submit?journalID=pgenetics&manu=PGENETICS-D-20-00662R1

**Press Queries**

---

## [Editor Report · Acceptance letter]

8 Oct 2020

PGENETICS-D-20-00662R1 

Mesenchyme-derived IGF2 is a major paracrine regulator of pancreatic growth and function 

Dear Dr Sandovici, 

We are pleased to inform you that your manuscript entitled "Mesenchyme-derived IGF2 is a major paracrine regulator of pancreatic growth and function" has been formally accepted for publication in PLOS Genetics! Your manuscript is now with our production department and you will be notified of the publication date in due course.

With kind regards,

Matt Lyles

PLOS Genetics

On behalf of:
